# Pollution trace gas distributions and their transport in the Asian monsoon upper troposphere and lowermost stratosphere during the StratoClim campaign 2017

Sören Johansson[1], Michael Höpfner[1], Oliver Kirner[2], Ingo Wohltmann[3], Silvia Bucci[4], Bernard Legras[4], Felix Friedl-Vallon[1], Norbert Glatthor[1], Erik Kretschmer[1], Jörn Ungermann[5], and Gerald Wetzel[1]

[1]Institute of Meteorology and Climate Research, Karlsruhe Institute of Technology, Karlsruhe, Germany
[2]Steinbuch Centre for Computing, Karlsruhe Institute of Technology, Karlsruhe, Germany
[3]Alfred Wegener Institute, Helmholtz Center for Polar and Marine Research, Potsdam, Germany
[4]Laboratoire de Météorologie Dynamique, UMR8539, IPSL, CNRS/PSL-ENS/Sorbonne Université/École polytechnique, Paris, France
[5]Institute of Energy and Climate Research - Stratosphere (IEK-7), Forschungszentrum Jülich, Jülich, Germany

**Correspondence:** S. Johansson (soeren.johansson@kit.edu)

**Abstract.** We present the first high resolution measurements of pollutant trace gases in the Asian Summer Monsoon Upper Troposphere and Lowermost Stratosphere (UTLS) from the Gimballed Limb Observer for Radiance Imaging of the Atmosphere (GLORIA) during the StratoClim (Stratospheric and upper tropospheric processes for better climate predictions) campaign based in Kathamandu, Nepal, 2017. Measurements of peroxyacetyl nitrate (PAN), acetylene ($C_2H_2$), and formic acid (HCOOH) show strong local enhancements up to altitudes of 16 km. More than 500 pptv of PAN, more than 200 pptv of $C_2H_2$, and more than 200 pptv of HCOOH are observed. Air masses with increased volume mixing ratios of PAN and $C_2H_2$ at altitudes up to 18 km, reaching to the lowermost stratosphere were present at these altitudes for more than 10 days, as indicated by trajectory analysis. A local minimum of HCOOH is correlated with a previously reported maximum of ammonia ($NH_3$), which suggests different wash out efficiencies of these species in the same air masses. A backward trajectory analysis based on the models ATLAS and TRACZILLA, using advanced techniques for detection of convective events, and starting at geolocations of GLORIA measurements with enhanced pollution trace gas concentrations has been performed. The analysis shows that convective events along trajectories leading to GLORIA measurements with enhanced pollutants are located close to regions, where satellite measurements by the Ozone Monitoring Instrument (OMI) indicate enhanced tropospheric columns of nitrogen dioxide ($NO_2$) in the days prior to the observation. A comparison to the global atmospheric models Copernicus Atmosphere Monitoring Service (CAMS) and ECHAM/MESSy Atmospheric Chemistry (EMAC) has been performed. It is shown that these models are able to reproduce large scale structures of the pollution trace gas distributions for one part of the flight, while the other part of the flight reveals large discrepancies between models and measurement. These discrepancies possibly result from convective events that are not resolved or parameterized in the models, uncertainties in the emissions of source gases, and uncertainties in the rate constants of chemical reactions.

# 1 Introduction

During the Asian Summer Monsoon (ASM), a large-scale persistent anticyclonic circulation, the so-called Asian Monsoon Anticyclone (AMA), exists in the upper troposphere. Convective processes are able to inject polluted lower tropospheric air masses into the AMA (Randel and Jensen, 2013; Vogel et al., 2015; Pan et al., 2016; Legras and Bucci, 2020). So far, most observational information of the Upper Troposphere and Lowermost Stratosphere (UTLS) composition during the ASM has been obtained by satellite limb-sounding experiments (e.g., Santee et al., 2017) which are, however, restricted by relatively coarse vertical and horizontal resolution and sampling. Airborne in-situ observations of air masses belonging to the AMA are extremely sparse and often sample only filaments, border areas, or outflow of the AMA (e.g., Bourtsoukidis et al., 2017; Gottschaldt et al., 2018). Here, we present data from the first dedicated aircraft campaign sampling air from the central region of the AMA.

In the contemporary understanding, the chemical composition of the confined system of the AMA is dominated by enhanced amounts of tropospheric trace gases, such as water vapor ($H_2O$) and carbon monoxide (CO), which are transported to higher altitudes (Santee et al., 2017). Several model studies have identified the impact of different source regions and long-range transport on the AMA, either with artificial tracers (e.g., Vogel et al., 2015, 2016, 2019) or e.g., CO as a proxy (Pan et al., 2016; Cussac et al., 2020). However, large discrepancies are observed in ASM radiative heating rates between different reanalyses (Randel and Jensen, 2013), which result in uncertainties in diabatic vertical transport. Together with large uncertainties in vertical transport due to convection (e.g., Hoyle et al., 2011), atmospheric models are limited in their ability to simulate air masses within the ASM. Comparisons of pollutant trace gas observations in the upper troposphere with simulation results may help to identify problems in atmospheric models.

It has been shown that pollutants, such as ammonia ($NH_3$) or peroxyacetyl nitrate (PAN) are first transported to the upper troposphere and then accumulated there (Höpfner et al., 2016; Ungermann et al., 2016). These pollutants play an important role in the formation of upper tropospheric ozone and aerosol (e.g., Singh, 1987; Höpfner et al., 2019). Airborne in-situ measurements of filaments with polluted monsoon air masses during the ESMVal (Earth System Model Validation) campaign have revealed that $O_3$ increases in the AMA outflow in tropospheric air (Gottschaldt et al., 2018). Based on in-situ measurements of another aircraft campaign (Oxidation Mechanism Observations), Lelieveld et al. (2018) formulated the picture of the AMA as a pollution pump and purifier by hydroxyl radicals (OH) of the polluted air masses, originating from South Asian emissions: Due to the production of OH by reactions with lightning nitrogen oxide ($NO_x$) in the upper troposphere, the polluted air masses are processed with the available OH. Besides these measurements of filaments and outflow air, observations of the ASM UTLS in high spatial resolution are sparse.

The first aircraft observations sampling air in the center of the AMA have been obtained during the high-altitude airborne StratoClim (Stratospheric and upper tropospheric processes for better climate predictions) campaign. This study presents a unique data set of pollution trace gases, in particular non-methane volatile organic compounds (NMVOCs) obtained with the Gimballed Limb Observer for Radiance Imaging of the Atmosphere (GLORIA) during this StratoClim campaign based in

Kathmandu, Nepal, 2017. One goal of the GLORIA measurements was to identify and quantify the spatial distribution of the pollution trace gases PAN, $C_2H_2$, and HCOOH, together with $HNO_3$ and $O_3$ in the Asian monsoon UTLS.

In this study, we use the NMVOC measurements from GLORIA collected during StratoClim to address two important science objectives. First, the origin of polluted air masses is investigated through two trajectory models allowing advanced schemes for detection of convective vertical transport times and areas. Second, a first evaluation in the ASM UTLS in high spatial resolution of the atmospheric chemistry models EMAC and CAMS is provided.

In the following section, methods and data sets are introduced: First, the discussed pollution trace gases are briefly characterized, the StratoClim aircraft campaign is described and the GLORIA measurements and data evaluation explained, and the atmospheric models applied here are introduced. Then, the GLORIA measurements are discussed (Section 3). This is followed by a presentation of the analysis of ATLAS and TRACZILLA backward trajectories from air masses of interest identified by the GLORIA measurements (Section 4). Section 5 presents a comparison to the atmospheric chemistry models EMAC and CAMS.

## 2 Data sets and methods

### 2.1 Measured trace gases

The scientific analysis of the trace gas measurements will be based on the five species $HNO_3$, $O_3$, PAN, $C_2H_2$, and HCOOH, which are briefly introduced in the following paragraphs. Focus is put on sources and sinks of these species, their lifetimes in the atmosphere, and reports of earlier measurements.

#### 2.1.1 Nitric acid

Nitric acid ($HNO_3$) is a trace gas with maximum mixing ratios of several ppbv in the stratosphere (e.g., Brasseur and Solomon, 2005). In the troposphere, $HNO_3$ typically has mixing ratios of less than 1 ppbv. Atmospheric $HNO_3$ acts as a sink of tropospheric $NO_x$ (e.g., $NO_2+OH\rightarrow HNO_3$) but is also part of a variety of other chemical reactions, which are not mentioned here in detail (see e.g., Burkholder et al., 2015). Tropospheric $NO_x$ may result from fossil fuel combustion, biomass burning, lightning, soil emissions, and stratospheric intrusions (Schumann and Huntrieser, 2007).

There is a large number of spaceborne $HNO_3$ observations reaching down into the upper troposphere, e.g., volume mixing ratio (VMR) profiles from the Microwave Limb Sounder (MLS; Santee et al., 2007, 2011), the Atmospheric Chemistry Experiments - Fourier Transform Spectrometer (ACE-FTS; Wolff et al., 2008), or the Michelson Interferometer for Passive Atmospheric Sounding on the Envisat satellite (MIPAS; von Clarmann et al., 2009). Airborne measurements have been performed in-situ using chemical ionization mass spectrometer techniques (e.g., Neuman et al., 2001; Jurkat et al., 2016), and also with remote sensing techniques (e.g., Braun et al., 2019). In the ASM, $HNO_3$ is used as a stratospheric tracer to demonstrate the isolation of the AMA (Park et al., 2008) and is used for studies of transport and chemistry of peroxyacetyl nitrate (PAN; Fadnavis et al., 2014). In this paper, $HNO_3$ measurements serve as complementary information and are not analyzed in detail.

### 2.1.2 Ozone

Ozone ($O_3$), similarly to $HNO_3$, has its maximum in the stratosphere, but with much higher mixing ratios of several ppmv (e.g., Brasseur and Solomon, 2005). Due to the importance of the stratospheric ozone layer, stratospheric $O_3$ is continuously monitored (WMO, 2019). In the ASM troposphere, $O_3$ has typical background values of 25-150 ppbv (e.g., Brunamonti et al.,
2018). Sources of tropospheric $O_3$ are stratosphere-troposphere exchange processes, and in situ production from $NO_x$ and hydrogen oxide radicals, and from $NO_x$ and peroxide radicals from oxidation of volatile organic compounds (Brasseur and Solomon, 2005; Bozem et al., 2017). Thus, tropospheric $O_3$ can be an indicator of polluted air masses. Furthermore, lightning $NO_x$ may increase tropospheric ozone. Tropospheric enhancements of $O_3$ typically can reach up to several hundred ppbv. Loss of tropospheric $O_3$ is caused by photolysis, reactions with OH, and reactions with $NO_x$ (e.g., Bozem et al., 2017). For that
reason, moist and clean air masses are correlated with rather low $O_3$ mixing ratios. During the Asian monsoon, typically low tropospheric $O_3$ mixing ratios are measured due to uplift of moist and clean air from the Indian Ocean (e.g., Safieddine et al., 2016; Santee et al., 2017; Brunamonti et al., 2018).

Measurements of $O_3$ in the upper troposphere are available e.g., VMR profiles from MLS (Froidevaux et al., 1994; Livesey et al., 2008), ACE-FTS (Sheese et al., 2016), MIPAS (von Clarmann et al., 2009), from airborne in-situ (e.g., Browell et al.,
1987; Zahn et al., 2012; Safieddine et al., 2016) and airborne remote sensing measurements (e.g., Browell et al., 1987; Woiwode et al., 2012). During recent Asian monsoon seasons, balloon-borne ozone sondes have been launched (e.g., Bian et al., 2012; Brunamonti et al., 2018; Li et al., 2018). Similarly to $HNO_3$, enhanced $O_3$, within the generally $O_3$-poor AMA upper tropospheric air, is either interpreted as indicator of stratospheric air (e.g., Park et al., 2007, 2008) or connected to uplift of $O_3$ precursor species of polluted air (Gottschaldt et al., 2017).

### 2.1.3 Peroxyacetyl nitrate

PAN is a secondary pollutant formed by the reaction of peroxyacetyl with nitrogen dioxide:

$$CH_3COO_2 + NO_2 + M \rightleftharpoons CH_3COO_2NO_2 + M \tag{R1}$$

Peroxyacetyl is a product of oxidation or photolysis of NMVOC, which are emitted from fuel combustion and biomass burning (Fischer et al., 2014). PAN is mainly destroyed by thermal decomposition to the starting species of reaction (R1), whereas
dry deposition and photolysis play a minor role at lower tropospheric altitudes (e.g., Fadnavis et al., 2014). In the upper troposphere instead, photolysis is the dominant loss process for PAN (e.g., Fadnavis et al., 2015). While the lifetime of PAN is very short at lower altitudes due to rapid thermal decomposition (1 h at temperatures of 298 K), it becomes progressively longer and reaches around 5 months at higher altitudes (at temperatures of 250 K). Typical background abundances of PAN in the upper troposphere are below 100 pptv (Glatthor et al., 2007). This makes PAN useful as a tracer for upper tropospheric
transport studies (Singh, 1987; Glatthor et al., 2007; Fischer et al., 2014).

PAN VMR observations are reported from the spaceborne instruments ACE-FTS (Coheur et al., 2007), MIPAS-Envisat (Glatthor et al., 2007; Wiegele et al., 2012), CRISTA (Cryogenic Infrared Spectrometer and Telescope for the Atmosphere;

Ungermann et al., 2016), and column information from IASI (Infrared Atmospheric Sounding Interferometer; Coheur et al., 2009). Airborne measurements were achieved from instruments with remote sensing (CRISTA-NF; Ungermann et al., 2013), and in-situ (e.g., Singh et al., 2001) measurement techniques. In the ASM, measurements of PAN have been applied to study transport and impact of polluted air masses (Fadnavis et al., 2014). Typically, PAN is strongly enhanced within the main part of the AMA (Ungermann et al., 2016).

### 2.1.4 Acetylene

Acetylene or ethyne ($C_2H_2$), a product of biofuel and fossil fuel combustion and biomass burning, has maximum tropospheric mixing ratios of a few ppbv. Typical background values for $C_2H_2$ in the tropics are below 75 pptv (e.g., Xiao et al., 2007; Wiegele et al., 2012). The reaction with OH is the major sink of $C_2H_2$ in the troposphere. Compared to PAN, $C_2H_2$ has a rather short lifetime of 2 weeks (Xiao et al., 2007). Measurements of $C_2H_2$ VMR profiles have been reported by e.g., ATMOS (Atmospheric Trace Molecule Spectroscopy; Rinsland et al., 1987), ACE-FTS (Rinsland et al., 2005), and MIPAS-Envisat (Wiegele et al., 2012). Within the ASM, measurements of $C_2H_2$ have been used as tropospheric tracer to study the chemical isolation of the AMA (Park et al., 2008) .

### 2.1.5 Formic acid

Formic acid (HCOOH) mainly exists in the troposphere (with background VMRs below 100 pptv and peak VMRs smaller than 1 ppbv (Grutter et al., 2010)) and originates from biogenic emissions, biomass burning, and fossil fuel combustion (Mungall et al., 2018). In contrast to PAN and $C_2H_2$, HCOOH is water soluble, such that its major tropospheric sink is wet deposition (depending on acidity). Further sinks are reaction with OH, and dry deposition (Paulot et al., 2011). The average atmospheric lifetime is estimated to be 2-4 days, while the lifetime is shorter in the boundary layer and longer in the free troposphere (Millet et al., 2015). Measurements of atmospheric HCOOH VMR profiles are available e.g., from ACE-FTS (Rinsland et al., 2006), MIPAS-Envisat (Grutter et al., 2010) and column information from IASI (Coheur et al., 2009). Airborne measurements in the UTLS are reported by e.g., Reiner et al. (1999) and Singh et al. (2000). To our knowledge, no studies using measurements of HCOOH in the ASM upper troposphere have been published so far.

### 2.2 StratoClim aircraft campaign and GLORIA observations

During the Asian summer monsoon 2017, the StratoClim aircraft campaign was conducted from Kathmandu, Nepal. In total, 22 in-situ and 3 remote sensing instruments were integrated into the Russian high-altitude research aircraft M55 Geophysica. As part of the remote sensing payload, GLORIA was deployed during four research flights of this measurement campaign. In the present work, we will discuss the research flight on 31 July 2017. This research flight was selected for this work due to high flight altitudes and low cloud top altitudes within the AMA, which both are optimal measurement conditions for the infrared limb sounding instrument GLORIA. This research flight was by far the best due to the flight length allowing different air masses to be sampled, and due to the low cloud top altitude. The related flight path is shown in Fig. 1.

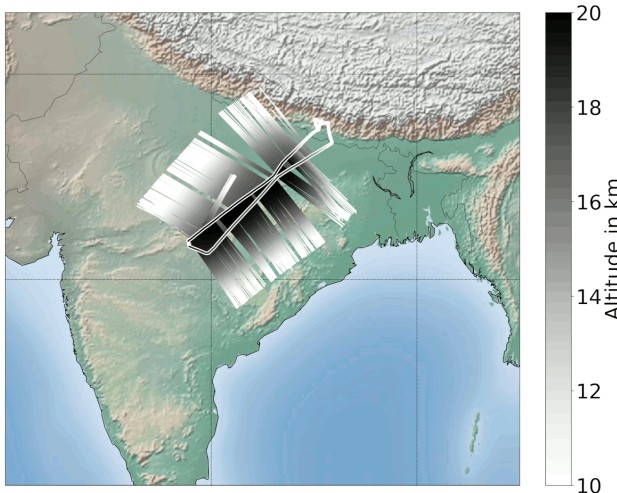

**Figure 1.** Flight path and tangent altitudes of StratoClim research flight on 31 July 2017. The bold line indicates the flight path and aircraft altitude of Geophysica, while small across-track lines indicate the positions and altitudes of tangent altitudes of GLORIA measurements. The map is centered at the Indian subcontinent.

GLORIA is a unique airborne imaging limb infrared sounding instrument (Friedl-Vallon et al., 2014; Riese et al., 2014), which has been operated during several campaigns with the German HALO research aircraft and with M55 Geophysica. The main part of GLORIA is a Michelson Fourier Transform Spectrometer combined with an imaging detector, which allows for simultaneous measurements of $127 \times 48$ spectra. Further, GLORIA consists of two external black bodies for in-flight radio-

5   metric calibration and a gimbal frame for active corrections of aircraft movements and line-of-sight control. Interferograms used in this study are from the "high spectral resolution" mode with 8.0 cm optical path difference, which results in spectra with $0.0625 \, \text{cm}^{-1}$ sampling.

These measured spectra are then used to retrieve profiles of atmospheric trace gases and particles. The retrieval is performed using a nonlinear least-squares fit with Tikhonov regularization. The overall retrieval strategy is explained in detail by Johans-

10   son et al. (2018). The retrieval strategy used here differs from that described by Johansson et al. (2018) mainly in the applied cloud filter and the handling of continua in the radiative transfer model KOPRA (Stiller, 2000): Due to the high mixing ratios of aerosols (Höpfner et al., 2019), cloud filtering using the MIPAS "cloud index" method (Spang et al., 2004) was replaced by filtering according to mean radiance between $850 \, \text{cm}^{-1}$ and $970 \, \text{cm}^{-1}$ with a threshold of $800 \, \text{nW} \, (\text{cm}^2 \, \text{sr} \, \text{cm}^{-1})^{-1}$. Due to highly structured distributions of aerosol and thin cirrus, the retrieval of a spectrally flat extinction was replaced by retrieval

15   of a multiplicative scale and an additive radiance offset parameter for each vertical point in the retrieval altitude grid. For the $HNO_3$ retrieval, a pre-fitted scale from the spectral region $955.8750 - 958.4375 \, \text{cm}^{-1}$ has been used and only the offset was fitted (instead of scale and offset). Spectral ranges for the retrievals for each discussed trace gas and the handling of interfering species are summarized in Tab. 1, together with typical vertical resolutions and estimated errors. Detailed plots of estimated errors and vertical resolutions are provided as supplement to this paper.

**Table 1.** Retrieval properties for $HNO_3$, $O_3$, PAN, $C_2H_2$, and HCOOH: Spectral regions used, and handling of interfering species. 10 and 90 percentile ranges are given for vertical resolution and estimated errors. In the supplement, it is shown that larger absolute errors are typically connected to higher VMRs.

| target gas | spectral regions in cm$^{-1}$ | fitted species | forward-calculated species | vertical resolution | estimated error |
|---|---|---|---|---|---|
| $HNO_3$ | 867.0000 - 870.0000 | $HNO_3$, $NH_3$, OCS | $H_2O^{\dagger}$ | 0.7 - 0.9 km | 70 - 220 pptv |
| $O_3$ | 780.6250 - 781.7500 | $O_3$ | $H_2O$, $CO_2$ | 0.5 - 1.3 km | 70 - 200 ppbv |
| | 985.0000 - 988.0000 | | | | |
| PAN | 780.3125 - 790.0000 | PAN, $H_2O$, $O_3$, $CCl_4$ | $CO_2$, $HNO_3^{\dagger}$, CFC-22, | 0.5 - 0.8 km | 50 - 120 pptv |
| | 794.0000 - 805.0000 | | CFC-113, $ClONO_2$, $HNO_4$ | | |
| $C_2H_2$ | 759.5625 - 759.8125 | $C_2H_2$, $O_3$ | $H_2O^{\dagger}$, $CO_2$, $NO_2$, $NH_3^{\dagger}$, | 0.7 - 1.0 km | 30 - 60 pptv |
| | 766.5625 - 766.8125 | | $HNO_3^{\dagger}$, HCN, $CH_3Cl$, $C_2H_6^{\dagger}$, | | |
| | 775.9375 - 776.3125 | | $COF_2$, CFC-22, $CCl_4$, $N_2O_5$, | | |
| | 780.5000 - 780.8750 | | $ClONO_2$, $CH_3CCl_3$, $PAN^{\dagger}$ | | |
| HCOOH | 1103.5000 - 1106.1250 | HCOOH | $H_2O^{\dagger}$, $O_3^{\dagger}$, $CH_4$ | 0.7 - 1.0 km | 30 - 70 pptv |
| | 1112.5000 - 1116.8750 | | CFC-11, CFC-12, CFC-113 | | |

$^{\dagger}$ Results of previous retrievals (not necessarily shown in this paper) targeting these species have been used for simulation of the spectra.

## 2.3 Atmospheric model simulations

### 2.3.1 EMAC

The ECHAM/MESSy Atmospheric Chemistry (EMAC) model is a numerical chemistry and climate simulation system that includes sub-models describing tropospheric and middle atmospheric processes and their interactions with oceans, land, and human influences (Jöckel et al., 2010). It uses the second version of the Modular Earth Submodel System (MESSy) to link multi-institutional computer codes. The core atmospheric model is the 5th generation European Centre Hamburg general circulation model (ECHAM5; Roeckner et al., 2006). For the present study, we applied EMAC (ECHAM5 version 5.3.02, MESSy version 2.53) in the T106L90MA resolution, i.e. with a spherical truncation of T106 (corresponding to a quadratic Gaussian grid of approx. 1.125° × 1.125° (latitude × longitude) with 90 vertical hybrid pressure levels of up to 0.01 hPa (approx. 80 km). For convection, we use the parameterization introduced by Tiedtke (1989) with modifications by Nordeng (1994) as described in Tost et al. (2006). The chemical setup of the chemistry submodel MECCA (Sander et al., 2011) was selected with focus on the simulation of PAN and tropospheric chemistry. It compromises 460 chemical substances, 1187 gas phase reactions, and 262 photolysis reactions. The boundary conditions in our simulation are similar as in the EMAC hindcast simulations within the ESCiMo project (Earth System Chemistry integrated Modelling; Jöckel et al., 2016) and CCMI project (Chemistry-Climate Model Initiative; Eyring et al., 2013), which among others were performed for the WMO report "Scientific

Assessment of ozone depletion" (WMO, 2019). The boundary conditions for the mixing ratios of the greenhouse gases are from the RCP 6.0 scenario (Representative Concentration Pathway; Meinshausen et al., 2011). For the emission of NMVOCs, a data set of the MACCity emission inventory (MACC/CityZEN; Granier et al., 2011) and a data set consisting of a combination of ACCMIP (Atmospheric Chemistry and Climate Model Intercomparison Project; Lamarque et al., 2013) and RCP 6.0 data

(Fujino et al., 2006) were considered. There are anthropogenic emissions sources from biomass burning, agricultural waste burning, fossil fuels, ship, road and aircraft emissions, as well as biogenic emissions. For the simulated year 2017, the emissions of the year 2010 (the most recent year available) are repeated. We performed a standard simulation and a sensitivity simulation with additional 50% emissions of NMVOC. Thereby, the emissions were increased globally in all NMVOC emission sources by 50%. In both simulations, the meteorological fields were specified by the European Centre for Medium-Range Weather

Forecasts (ECMWF) European Reanalysis Interim (ERA-Interim; Dee et al., 2011), and all simulations were initialized on 1 May 2017.

### 2.3.2   CAMS

The Copernicus Atmosphere Monitoring Service (CAMS) reanalysis (CAMSRA) of ECMWF is an atmospheric composition model focusing on the troposphere. CAMS applies the ECMWF IFS (Integrated Forecast System) model and assimilates var-

ious satellite measurements of atmospheric composition. It uses the chemistry module IFS(CB05) (Flemming et al., 2015) and the aerosol module as described in Morcrette et al. (2009). Apart from assimilated ozone, CAMS does not simulate stratospheric chemistry. The model uses 60 vertical hybrid pressure levels, with the top level at 0.1 hPa and has a horizontal resolution of 80 km. Output is provided every 3 hours. The data set is characterized in detail by Inness et al. (2019). Anthropogenic emissions are prescribed by MACCity (Granier et al., 2011), biogenic emissions by MEGAN2.1 (Model of Emissions of Gases and

Aerosols from Nature; Guenther et al., 2012), and biomass burning emissions by GFAS v1.2 (Global Fire Assimilation System; Kaiser et al., 2012). CAMS reanalysis data is available for the time between 2003 and 2018. In a model evaluation study by Wang et al. (2020), trace gas measurements from aircraft campaigns were compared to CAMS reanalysis data. Tropospheric profiles (up to 12 km) of $O_3$, $HNO_3$, and PAN above Hawaii showed an agreement within the uncertainties of measurement and model. These agreements encourage model evaluation at altitudes of the upper troposphere in the ASM, as presented in

this study.

### 2.3.3   TRACZILLA

The TRACZILLA model (Pisso and Legras, 2008) is a modified version of the Lagrangian model FLEXPART (Stohl et al., 2005; Legras et al., 2005). Trajectories are launched at GLORIA tangent points at a rate of 1000 per point. They are integrated backward in time for one month, using the ECMWF reanalysis horizontal winds (European Reanalysis 5, ERA5, 1 h temporal

resolution), and diabatic vertical motions. The TRACZILLA simulations are run on a $0.25° \times 0.25°$ (latitude $\times$ longitude) grid at 137 vertical levels in the spatial domain of $10°W$ to $160°E$, and $0°N$ to $50°N$.

A diffusion with a diffusion coefficient of D=0.1 $m^2s^{-1}$ is added (based on previous studies in the subtropics, Pisso and Legras, 2008; James et al., 2008), represented by a random walk equivalent that disperses the cloud of parcels from each

point. A trajectory is considered to be influenced by convection if it encounters a convective cloud during its advection with a pressure higher than the cloud top pressure (as similarly done by Tissier and Legras, 2016). The location of cloud encounter is then identified as a convective source. We use here the cloud top products provided by the SAF-NWC (EUMETSAT Satellite Application Facility for Nowcasting) software package (Derrien et al., 2010; Sèze et al., 2015) from MSG1 (Meteosat 8) and Himawari geostationary satellites. More details on the algorithm and its evaluation against trace gas measurements can be found in Bucci et al. (2020) and Legras and Bucci (2020).

### 2.3.4 ATLAS

Trajectories from the Alfred Wegener InsTitute LAgrangian Chemistry/Transport System (ATLAS) model (Wohltmann and Rex, 2009) are driven by the same ECMWF ERA5 meteorological fields as TRACZILLA, but with a temporal resolution of 3 h and a horizontal resolution of $1.125° \times 1.125°$. The model uses a hybrid coordinate transforming from pressure at the surface to potential temperature at the tropopause (Wohltmann and Rex, 2009). In the altitude range of GLORIA, the coordinate is a potential temperature coordinate in good approximation, and the trajectories can be regarded as diabatic trajectories driven by the total heating rates of ERA5. Trajectories are calculated for 30 days prior to the measurement. Time step is 10 minutes.

The trajectory model includes a detailed stochastic parameterization of convective transport driven by ERA5 convective mass fluxes and detrainment rates (Wohltmann et al., 2019). In addition, a vertical diffusion of D=0.1 $m^2s^{-1}$ is added to every trajectory, consistently with TRACZILLA. At every measurement location of GLORIA, 1000 backward ensemble trajectories are started, which take different paths due to the stochastic nature of the convective transport scheme. A convective event is detected by drawing a uniformly distributed random number between 0 and 1 every 10 minutes and comparing that to the calculated probability for detrainment from ERA5 at the location of the trajectory. If the random number is smaller than the calculated probability, it is assumed that a convective event was encountered.

In an analysis of the ATLAS trajectories, the influence of the usage of ERA5 or ERA-Interim as meteorological fields, the influence of applied vertical diffusion, and the influence of the usage of kinematic or diabatic trajectories were investigated (shown in Fig. 19 of the Supplementary Information). This analysis (and also similar analyses by Legras and Bucci (2020)) revealed that major differences occur between ATLAS trajectories that use ERA5 or ERA-Interim meteorological fields. These major differences are exemplarily visible in Supplementary Fig. 19, where trajectory paths and locations of convective events are considerably different between ERA-Interim and ERA5. Compared to these large discrepancies, differences in trajectory paths and locations of convective events due to the usage of kinematic or diabatic trajectories, or due to the application of vertical diffusion are small.

### 2.4 OMI NO$_2$ tropospheric column

The Ozone Monitoring Instrument (OMI) is a nadir looking ultraviolet-visible spectrometer onboard the NASA Aura satellite. Targeted quantities are columns of trace gases (such as $O_3$, $NO_2$, $SO_2$), aerosol properties, cloud top heights, and surface irradiances (Levelt et al., 2006). In this work, we use the tropospheric column of nitrogen dioxide ($NO_2$) version 3 as a proxy for pollutant emissions (Crutzen, 1979). OMI tropospheric column $NO_2$ is provided as cloud-filtered daily level-3 product on

a 0.25° × 0.25° (latitude × longitude) grid (Krotkov, 2013). The version 3 standard retrieval of tropospheric column $NO_2$ comes with a spatial resolution of 1.0° × 1.25° (latitude × longitude), and showed an overall agreement with other satellite and ground-based measurements of $NO_2$ (Krotkov et al., 2017). For the analysis in Sec. 4, 14 days of OMI $NO_2$ measurements are averaged to indicate regions of pollutant emissions prior to the measurement.

## 3 Pollution trace gases measured during StratoClim flight on 2017-07-31

GLORIA measurements of pollution trace gases introduced in Sec. 2.1 are shown as horizontal distributions on a map, and as vertical distributions in Fig. 2, and air masses that are discussed in this and following sections are marked with colored boxes. As expected, the $HNO_3$ VMRs (Fig. 2b) strongly increase a few km above the tropopause, at 19 km to typical stratospheric values of more than 2 ppbv. Around the tropopause at 17 km, fluctuations of $HNO_3$ up to 0.5 ppbv can be observed. In the first part of the flight (until 4:45 UTC), also a local maximum of VMRs up to 1.0 ppbv is visible below the tropopause at altitudes between 15.5 km and 17 km (close to the red box in Fig. 2b). This maximum is continued by enhancements noted at 16 km at 4:00 UTC moving down to 15 km at 4:15-4:50 UTC with VMRs up to 0.75 ppbv (marked with a magenta box). The shape and position of the red and magenta boxes are optimized for the pollution trace gases PAN and $C_2H_2$ discussed later in this section to have a local maximum in the red and a local minimum in the magenta box. Thus, these boxes do not exactly match the structure in $HNO_3$. In addition, Höpfner et al. (2019) reported enhanced ammonium nitrate abundances in the red air masses, and a local minimum of ammonium nitrate in the magenta box. Given these different pollution trace gas and aerosol concentrations in the red and magenta boxes, it is assumed that these air masses have different origins, even though the structure in $HNO_3$ appears to be connected. In the second flight part, at 17 km altitude and 5:45 UTC, a local maximum of 0.5 ppbv $HNO_3$ is visible, together with a local minimum above, at 18 km altitude.

Measured $O_3$ (Fig. 2d) shows similar distributions as $HNO_3$: Above 19 km altitude, elevated VMRs of more than 1000 ppbv are measured in the same region where the $HNO_3$ measurements show stratospheric values. Above and around the 380 K tropopause, $O_3$ mixing ratios between 200 ppbv and 400 ppbv are measured. During the second part of the flight, at around 5:30 UTC, a local maximum of $O_3$ up to 400 ppbv was observed between 15 km and 16.5 km (purple box), but these measurements have relatively high total estimated errors up to 160 ppbv, which is within the magnitude of this local enhancement (see Supplementary Fig. 4).

PAN vertical distributions (Fig. 2f) show how different the air masses sampled with GLORIA have been: The first part (until 4:45 UTC) indicates enhanced PAN mixing ratios of more than 500 pptv up to the flight altitude of 18 km. Maximum values are observed at 4:10 UTC at 16 km altitude (marked with a red box). A PAN local minimum can be found in the same region of the $HNO_3$ local maximum (magenta box). These retrieved structures come with larger uncertainties compared to other parts of this flight, due to stronger spectral extinction observed at these altitudes in this part of the flight. The rather broadband spectral signature of PAN makes it more difficult for the retrieval to distinguish between the spectral offset due to aerosol or cirrus clouds, and the targeted spectral emission of PAN. The second part of the flight shows PAN background mixing ratios of 150 pptv in the troposphere and 100 pptv in the stratosphere. At lower altitudes, below 14 km, again enhanced mixing ratios of

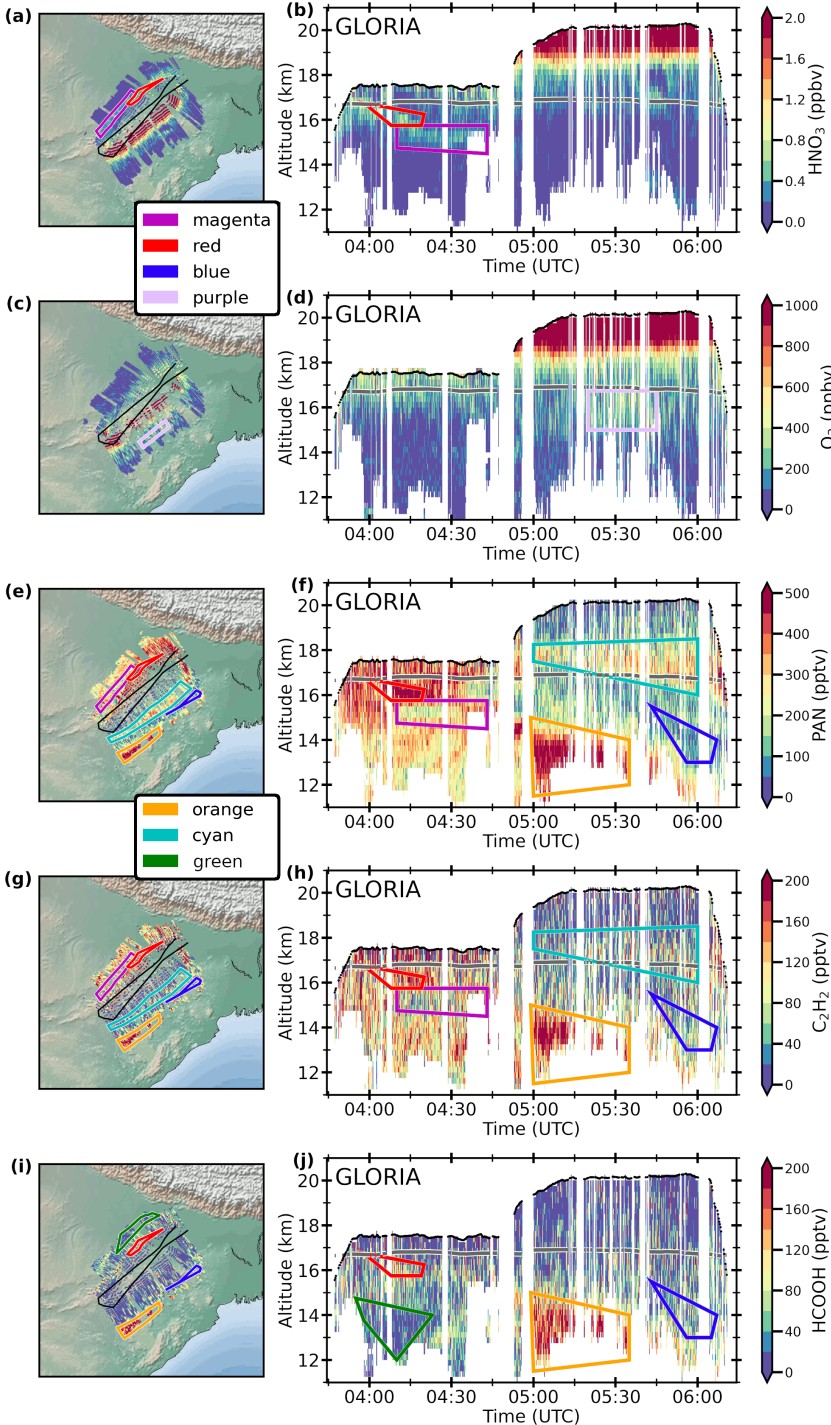

**Figure 2.** Horizontal (left column) and vertical (right column) distributions of GLORIA measurements of (a-b) $HNO_3$, (c-d) $O_3$, (e-f) PAN, (g-h) $C_2H_2$, (i-j) HCOOH for StratoClim flight on 31 July 2017. The black line in the maps and cross sections shows the flight path, and the dark gray line (on the cross section plots only) the 380 K potential temperature isentrope as approximate tropopause altitude. Colored boxes mark air masses of interest, discussed in the following sections. The maps are centered at the Indian subcontinent.

more than 500 pptv are observed between 5:00 and 5:30 UTC (orange box). Later during this flight, at 6:00 UTC, background values are observed at similar altitudes (blue box). Directly at and above the thermal tropopause, a local enhancement of 250 - 400 pptv of PAN is measured in this second part of the flight (cyan box).

Similarly to PAN, vertical distributions of $C_2H_2$ (Fig. 2h) show a large vertical variability during the first part of the flight. A local maximum of up to 200 pptv is observed at 4:10 UTC at 16 km altitude (marked with a red box) and a local minimum is visible below (magenta box). Below 14 km altitude, VMRs between 100 pptv and 200 pptv are measured. The second part of the flight shows strong enhancements of $C_2H_2$ of more than 200 pptv at altitudes below 14 km between 5:00 and 5:30 UTC (orange box), where PAN showed enhancements, too. The same local minimum as for PAN is observed at similar altitudes but later during the flight (6:00 UTC, blue box). Above the tropopause (cyan box), there is a minor local maximum with $C_2H_2$ VMRs of up to 100 pptv.

Unlike PAN and $C_2H_2$, HCOOH does not have these enhanced VMRs at and above the tropopause in the second part of the flight. Enhancements of HCOOH are observed in the first part of the flight at 4:10 and 16 km altitude (red box) and at 4:20 and 13 km altitude. In the second part of the flight, as for all gases other than $HNO_3$ and $O_3$, considerably larger abundances of HCOOH of more than 200 pptv are observed below 14 km between 5:00 and 5:30 UTC (orange box) and a local minimum is measured at similar altitudes at 6:00 UTC (blue box). At the beginning of the flight (until 04:15 UTC), below 15 km, a minimum of HCOOH is measured, while PAN and $C_2H_2$ are present (green box). This is the same air mass, where Höpfner et al. (2019) reported strongly enhanced mixing ratios of $NH_3$. This HCOOH local minimum is present, despite enhancements of HCOOH total columns (measured by IASI) at the border region between Pakistan and India (see Suppl. Fig. 11), the region identified as source for the enhanced $NH_3$. The trajectory analysis by Höpfner et al. (2019) indicated transport times of 3 days since convection, which is within the atmospheric lifetime of HCOOH. The presence of PAN and $C_2H_2$, together with the absence of HCOOH, suggests that the loss of HCOOH was induced by wet deposition, which was more efficient for HCOOH than for $NH_3$. Both species, HCOOH and $NH_3$, are water soluble, but HCOOH has a considerably higher Henry's Law Constant compared to $NH_3$. In addition, the solubility is known to be also dependent on the pH of the liquid (see e.g., Seinfeld and Pandis, 2016). HCOOH is acidic, in contrast to $NH_3$, which is alkaline. This difference is a possible explanation for HCOOH being washed out, while $NH_3$ is still present in large VMRs in the same air masses. However, we cannot rule out other processes leading to the different behaviors of HCOOH and $NH_3$ upon transport to the UT, like a difference in retention of these gases upon freezing of the liquid water droplets at high altitudes (Ge et al., 2018). As mentioned above, later during this flight, also strongly enhanced concentrations of HCOOH are observed. This indicates that air masses lifted by convection to high altitudes are not always depleted of HCOOH by washout processes.

The GLORIA measurements show how different the two parts of the flight have been regarding the composition of the measured air masses: In the first part of the flight, high VMRs of all discussed tropospheric tracers have been detected up to the tropopause at 17 km, while during the second part of the flight, such high abundances have been only detected at altitudes of up to 14 km. This difference between the flight legs is also discussed by Höpfner et al. (2019), who found large abundances of ammonia ($NH_3$) in the first part of the flight, but none in the second. On the map projections, it can be seen that the tangent points of the measurements point in opposite directions for the different flight parts: Lines of sight of the outbound flight leg

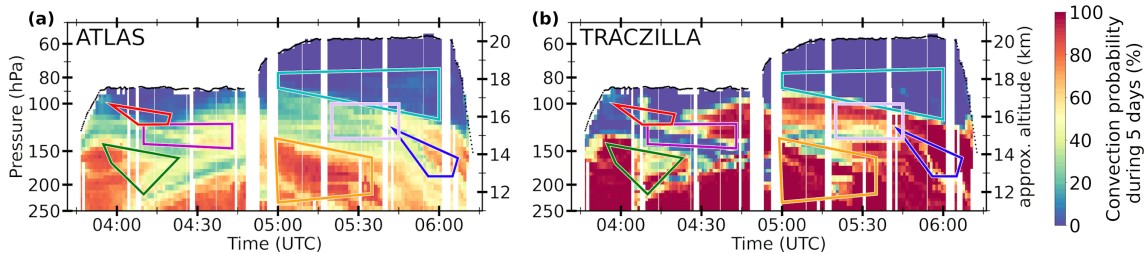

**Figure 3.** Convection probability of trajectories of the (a) ATLAS and (b) TRACZILLA model starting at the tangent points of the GLORIA measurements. The colors indicate the fraction of trajectories that experienced convection during the five days before the measurement time. Please note that the left vertical axis is shown as pressure in logarithmic scale. The right vertical axes display an approximation of altitude to facilitate the comparison with cross section plots in Fig. 2. Colored boxes are repeated from the same figure.

point towards north-west and those of the inbound flight leg point towards south-east. In addition, trajectory calculations by Höpfner et al. (2019) suggest that air masses observed during the first part of the flight were strongly influenced by convection a few days before the observations. Below 15 km, the beginning of the second part of the flight shows strong enhancements of pollution trace gases, and also an indication of convective events (which is discussed later in this paper). The enhancements

at and above the tropopause in the second flight part (cyan box) in PAN and $C_2H_2$, but not in HCOOH, suggest that these air masses are older than a few days (lifetime of HCOOH), but younger than 2 weeks (lifetime of $C_2H_2$). A detailed analysis of the origin of measured air masses of interest (marked by colored boxes) is discussed in Sec. 4.

## 4    Trajectory analysis: Origin of polluted air masses

To gain more insight about the origin of polluted air through convective processes, we apply backward trajectories from

the models TRACZILLA and ATLAS to identify the origin of measured polluted air masses. Due to the strong influence of convection, which is usually not resolved in the reanalysis data used by the models, both models use advanced methods for the detection of convective events along the trajectories (see Secs. 2.3.3-2.3.4). Fig. 3 shows the fraction of trajectories for each GLORIA measurement time and location that experienced convection during the past 5 days. This fraction is interpreted as convection probability and aids the following interpretation of the origin of polluted air masses. The ATLAS trajectories

indicate an enhanced convection probability of more than 50% for the orange and blue marked air masses, and parts of the purple and magenta marked air masses also have convection probabilities of 50%, while the red and cyan marked air masses show no noticeable enhancement of convection probability. For the TRACZILLA trajectories, the convection probabilities are overall higher. As for ATLAS, the orange and blue air masses have highest convection probabilities, the purple and magenta air masses have parts with high convection probability, and the red marked air masses have relatively low convection probabilities.

Another air mass of enhanced convection probability below 15 km ($\approx$140 hPa) and earlier than 04:10 UTC is present in both models (green box). This is the part of the flight, where Höpfner et al. (2019) report strongly enhanced ammonia VMRs. In their paper, the same sets of trajectories are used to estimate the source of origin at the border region between Pakistan and

India. The different absolute percentages for convection probability for ATLAS and TRACZILLA are likely the result of the different underlying data sets and different methods for detection of convection along the backward trajectories by the models.

In Fig. 4, we show the origin of the air masses in the colored boxes. As an example, Fig. 4a shows some exemplary 10-day backward trajectories from the ATLAS model started in the orange box (blue lines). In addition, the orange contours in the map show the spatial density of the convective events experienced by the backward trajectories originating in the orange box in the last 10 days. For this, all ensemble trajectories started in the orange box were taken into account (this comprises several GLORIA measurement locations with 1000 ensemble trajectories at each location). Now, the fraction of these trajectories that experienced a convective event in a given $1° \times 1°$ longitude $\times$ latitude box is calculated for each location on the map. The resulting quantity is a fraction per area, given here in units of percent-per-square-degree. The outermost contours correspond to a density of 0.1 percent of the started trajectories per square degree. In case that one or two inner contours are present, they correspond to 1 and 10 percent of the started trajectories per square degree, respectively.

To avoid confusion, note that Fig. 3 shows the fraction of the 1000 ensemble trajectories started from each individual GLORIA measurement location that experienced convection in the last 5 days as a function of this measurement location, which is not related to the location of the convective events or grouped by the colored boxes. Fig. 4b shows the same spatial densities as Fig. 4a, but now for all colored boxes and not only for the orange box. Fig. 4c shows the same as Fig. 4b for TRACZILLA. In addition, the OMI $NO_2$ tropospheric column is shown as an average over 14 days prior to the measurement as a proxy for emission of polluted air masses. These averaged tropospheric $NO_2$ columns are shown as background in green colors in Fig. 4b and c.

The ATLAS and TRACZILLA models show partly different areas of convective events for the labeled air masses of interest. A summary is given in Tab. 2. In the following parts, the plausibility of the calculated areas of convective events is tested by comparing these regions to the OMI $NO_2$ tropospheric column:

– The red air mass of interest has been selected due to enhanced pollution trace gas and $HNO_3$ measurements during the first part of the flight. For ATLAS and TRACZILLA, it was shown that only a small part of the trajectories experienced convection during 5 days before the measurement (see Fig. 3). In Fig. 4b-c, for both models regions are marked red between the location of the measurement and eastern China. For most regions marked red, convective influence along the trajectories was weak, thus only the 0.1 percent-per-square-degree contours are present. However, most regions marked red in northeastern China lie close to areas with enhanced $NO_2$, so these regions may possibly have contributed to the measured enhanced pollution trace gases. For the ATLAS model, it is shown in Supplementary Fig. 12 that for the red air mass, less than 30% of all started trajectories experienced a convective event within 10 days before the measurement, showing the weak convective influence.

– The magenta air mass of interest is connected to a local minimum of PAN and $C_2H_2$, and enhanced $HNO_3$, close to the maximum of the pollutant species marked with the red box. Both trajectory models show similar convective densities as for the red regions above China, and they also show substantial convective activity above the South China and Philippine Seas. TRACZILLA also indicates regions of strong convection northwest of the flight path, while ATLAS indicates a

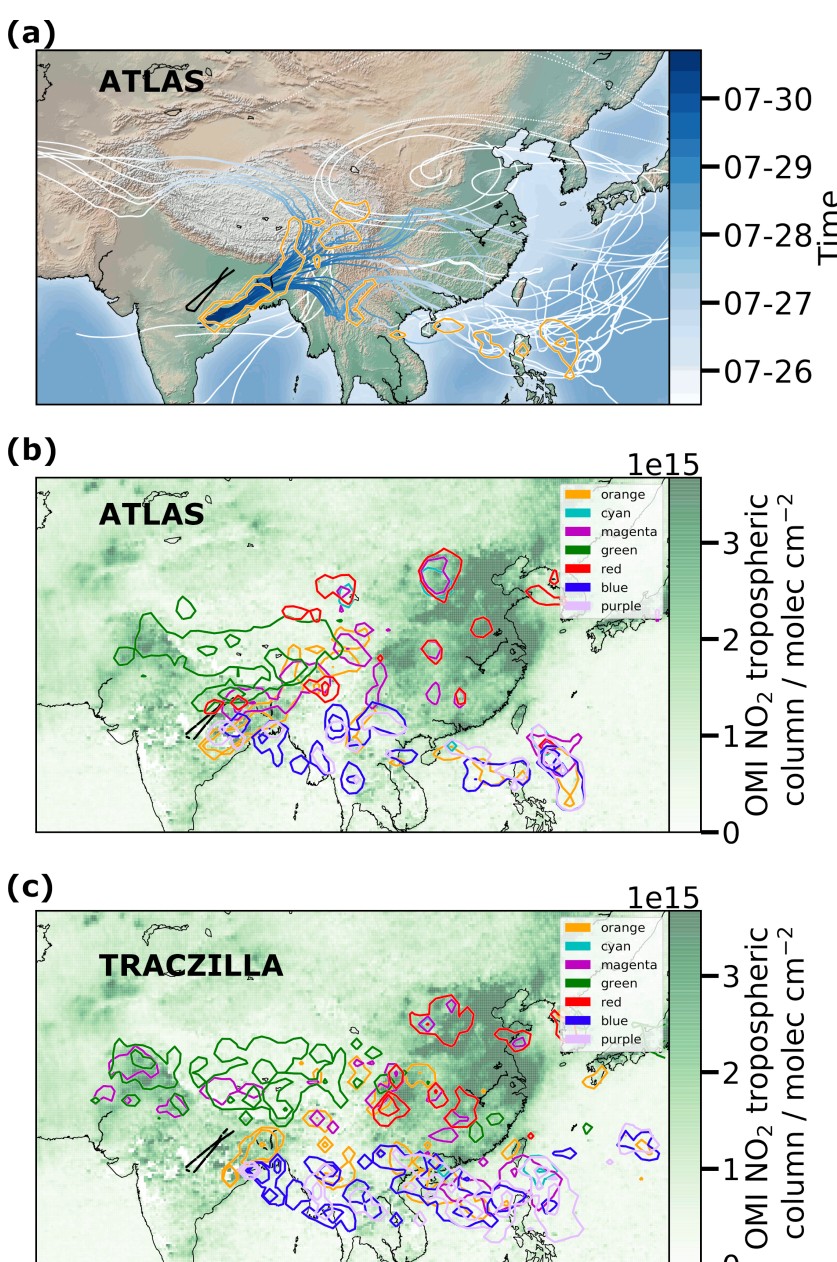

**Figure 4.** The origin of air masses of interest along the GLORIA vertical cross sections. (a) Exemplary subset of backward ATLAS trajectories starting at the air masses marked orange in Figs. 2-3. Backward trajectories are displayed in this figure for 10 days, of which the temporal evolution for the first 5 days is color coded according to the color bar. Orange colored regions on the map mark regions, where the density of convective events (as explained in Sec. 2.3.4) is larger than 0.1, 1.0, and 10.0 percent-per-square-degree (calculated as the number of encountered convective events over the number of total released trajectories; on a $1°$ latitude $\times$ $1°$ longitude grid). (b) Same regions as in (a), but for all colored air masses from Figs. 2-3. Background: OMI satellite measurements of the tropospheric $NO_2$ column, averaged over 14 days before the measurement. (c) Same as (b), but with density of convective events (as explained in Sec. 2.3.3) from TRACZILLA.

**Table 2.** Summary of air masses of interest marked with colored boxes (see Fig. 2), their altitude ranges, and local minima (↓) and maxima (↑) of trace gases measured by GLORIA (repeated from Sec. 3). Regions of origin, as indicated by ATLAS and TRACZILLA (see also Fig. 4) are listed. Approximate ages of air are shown in Supplementary Tab. 1.

| color | altitude | GLORIA measurements | ATLAS | TRACZILLA |
|---|---|---|---|---|
| red | 15.75-16.75 km | $HNO_3$: 1.0 ppbv ↑<br>PAN: >500 pptv ↑<br>$C_2H_2$: 200 pptv ↑<br>HCOOH: 150 pptv ↑ | north eastern India,<br>eastern China,<br>central China | eastern China<br>central China |
| magenta | 14.5-15.75 km | $HNO_3$: 0.75 ppbv ↑<br>PAN: 150 pptv ↓<br>$C_2H_2$: 100 pptv ↓ | eastern China,<br>South China and<br>Philippine Seas,<br>north eastern<br>India | eastern China,<br>South China and<br>Philippine Seas,<br>Tibetan Plateau<br>and Kashmir |
| orange | 11.5-15.0 km | PAN: >500 pptv ↑<br>$C_2H_2$: >200 pptv ↑<br>HCOOH: >200 pptv ↑ | eastern India,<br>southern China,<br>South China Sea | eastern India,<br>southern China,<br>South China Sea |
| blue | 13.0-15.5 km | PAN: <150 pptv ↓<br>$C_2H_2$: <80 pptv ↓<br>HCOOH: <60 pptv ↓ | Bay of Bengal,<br>Myanmar,<br>South China and<br>Philippine Seas | Bay of Bengal,<br>Myanmar,<br>South China and<br>Philippine Seas |
| purple | 15.0-16.75 km | $O_3$: 400 ppbv ↑ | eastern India,<br>South China and<br>Philippine Seas | eastern India,<br>South China and<br>Philippine Seas |
| cyan | 16.0-18.5 km | PAN: 400 pptv ↑<br>$C_2H_2$: 100 pptv ↑ | South China and<br>Philippine Seas,<br>central China | South China and<br>Philippine Seas |
| green | 12.0-14.75 km | HCOOH: <20 pptv ↓ | Tibetan Plateau,<br>Kashmir | Tibetan Plateau,<br>Kashmir |

region in north eastern India, close to the flight path. Again, for almost all these marked regions, convective influence on the measured air masses was weak, thus only the 0.1 percent-per-square-degree lines are present. However, in this case, it is likely that convection in the regions above the South China and Philippine Seas brought up clean maritime air. Enhanced $HNO_3$ concentrations within these air masses possibly result from reaction of lightning $NO_x$ with OH to $HNO_3$ (see e.g., Schumann and Huntrieser, 2007).

– The origin of the green colored air mass has been already discussed by Höpfner et al. (2019) on basis of similar trajectory sets and visualizations. Here, also regions with densities of convective events as low as 0.1 percent-per-square-degree

are shown, and regions larger than those described by Höpfner et al. (2019) are marked. However, still both trajectory models highlight regions of the Tibetan Plateau and Kashmir as possible sources of the green air masses of interest. While GLORIA HCOOH shows a local minimum, enhancements of $C_2H_2$, PAN, and $NH_3$ (not shown) are measured. This chemical composition of the green marked air masses makes an origin from the relatively clean Tibetan Plateau unlikely. In addition, Höpfner et al. (2019) connected a boundary layer maximum of $NH_3$ in the Kashmir region (measured by IASI) with these air masses, which makes this Kashmir region the most likely origin of the measured air masses marked green. A connection of the minimum of GLORIA HCOOH with IASI boundary layer measurements of HCOOH has been discussed in Sec.2.1.

– The second part of the flight has again an air mass with enhanced NMVOC measurements, marked with orange color. Both models indicate that the trajectories encountered convection not too far away from the flight path. Between the flight path and the coast, the 1 percent-per-square-degree, and for TRACZILLA even the 10 percent-per-square-degree convective density lines are visible. This region is connected to enhanced OMI $NO_2$ measurements, which is a strong indication for local emissions from India. Smaller regions marked orange in both models, present above southern China and the South China Sea, are not likely to contribute to the strongly enhanced measured pollutants. For the ATLAS model, it is shown in Supplementary Fig. 14 that for backward trajectories from the orange air mass, about 50% of all convective events occur spatially very close to the measurement location and within 3 days before the measurement. This corresponds to the orange region in India with enhanced $NO_2$ columns.

– A local minimum of NMVOCs at the same altitude of measurement as the air mass outlined in orange is marked blue. Both models indicate convective source regions between the flight path and the Bay of Bengal, together with regions above Myanmar, and the South China and Philippine Seas. All these regions show low $NO_2$ and are therefore plausible source regions for the rather clean air masses measured by GLORIA.

– The local maximum of $O_3$ in the second part of the flight below the tropopause is marked purple (see Fig. 2). Based on the convection along the model trajectories, the origin of these air masses is very similar to that of the blue labeled air masses: Convection probabilities along the trajectories from the purple air mass are enhanced over eastern India and above the South China and Philippine Seas. These areas marked by the trajectories show low OMI $NO_2$ and indicate relatively clean boundary layer air, which cannot explain the measured local enhancement of $O_3$. This suggests that the measured local maximum of $O_3$ is of other than convective origin; possibly, the measured maximum is a pollution remainder transported for more than 10 days, or an intrusion of stratospheric air.

– The cyan colored air masses are connected to the measured enhancement of PAN and $C_2H_2$ above the tropopause. In Sec. 3, it is suggested that these air masses are transported for more than a few days, but for less than two weeks. For this reason, it is not expected to see strong convective influence on the trajectories a few days prior to the measurement. This is also supported by Fig. 3, which does not show strongly enhanced convection probabilities in the cross sections for either trajectory model. ATLAS and TRACZILLA both only show a small convective region over the Philippine Sea for

the cyan air mass of interest, and ATLAS also shows convective activity above central China. For the ATLAS model, it is shown in Supplementary Fig. 17 that for backward trajectories from the cyan air mass, less than 20% of all trajectories experienced any convective events, which also indicates the low convective influence.

In summary, both models suggest, that air masses of interest with enhanced pollution trace gas measurements (orange, red) originate from India or China, where boundary layer pollution is also observed by OMI. Air masses of interest with low pollution trace gases measured are transported from maritime and pristine surface regions (blue). For air masses with low pollution trace gases, but locally enhanced $O_3$ or $HNO_3$ (magenta, purple), also maritime regions are indicated, which can explain low pollution trace gases, but not the enhanced $O_3$ or $HNO_3$ VMRs. As expected, the local maxima of PAN and $C_2H_2$ at altitudes of the tropopause (cyan) were transported for a longer time, according to the trajectory models. The origin of the local minima of HCOOH (green) already has been discussed to be most likely from the Kashmir region (Höpfner et al., 2019).

The comparison of ATLAS and TRACZILLA calculations of convective origin of the measured pollution species shows that there are few differences between these model results. Both models give results for the source regions and convective age of air that are broadly consistent with the measurements. Due to the numerous uncertainties, there are, however, also some results which seem to be less plausible. However, OMI $NO_2$, which is shown as proxy for boundary layer pollution, does not account for biogenic sources and is shown as average over 14 days prior to the measurement (see Sec. 2.4). Due to these limitations, additional pollution sources may have been overlooked in this analysis. Still, similar origins of highly polluted air masses, indicated by two independent backward trajectory models, agree with enhanced surface pollution, measured by OMI. This agreement within the anticipated accuracy of the two backward trajectory models suggests that both models use reliable schemes for convection detection.

## 5   Comparison to model simulations

Another application of our highly resolved measurements is shown in this section, in the evaluation of atmospheric models. Simulation results of the CAMS reanalysis and EMAC atmospheric models are interpolated onto the geolocations of the GLORIA measurements (see Fig. 1). Comparisons with the GLORIA measurements for each gas are shown in Fig. 5. Due to the rather coarse horizontal resolution of the models, compared to the distance between GLORIA profiles, the GLORIA cross sections for these model comparisons have been averaged over 33 single profiles, i.e. a horizontal distance of approx. 100 km.

Comparisons of GLORIA $HNO_3$ with CAMS (Fig. 5a-b) show that no noticeable enhancements are simulated by the CAMS model. $HNO_3$ data are provided in the CAMS reanalysis data product, but the model does not include stratospheric chemistry and thus no stratospheric $HNO_3$. The comparison of the observations with EMAC $HNO_3$ (Fig. 5c) shows several similarities: High stratospheric VMRs up to 2 ppbv are simulated above 19 km in the second part of the flight; they decrease to values of 0.5 ppbv at the tropopause. Simulated maximum stratospheric values are not always as high as the values measured, but they agree to within the GLORIA estimated error (see Supplement, Figs. 1 and 2). Fine-scale structures around the tropopause in the measurements are not reproduced by the model, which can be explained by the lower horizontal resolution of EMAC. Below the tropopause, the diagonal structure in the first part of the flight (of which parts are in the red and magenta boxes) is

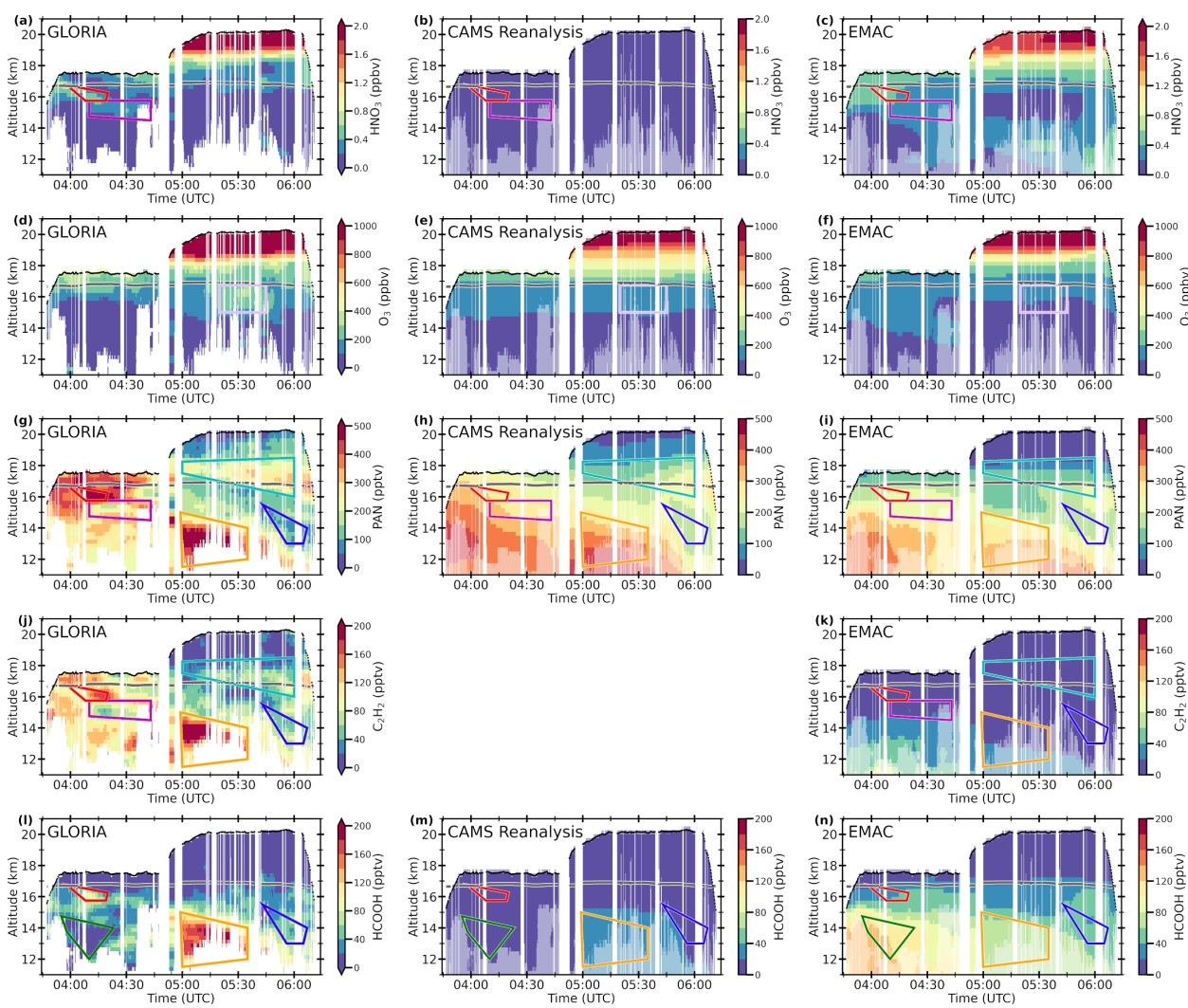

**Figure 5.** GLORIA (left column, similar to Fig. 2, but averaged over 33 single profiles), CAMS (middle column), and EMAC (right column) distributions of (a-c) $HNO_3$, (d-f) $O_3$, (g-i) PAN, (j-k) $C_2H_2$, (l-n) HCOOH for StratoClim flight on 31 July 2017. CAMS reanalysis does not include $C_2H_2$. All auxiliary lines as defined in the caption of Fig. 2. Altitudes not measured by GLORIA are marked with a white shadow in the model data.

reproduced by EMAC. In the model, this structure continues into the second part of the flight, which has not been observed by GLORIA. This diagonal structure coincides with the gradient of EMAC PAN (see below), which suggests that this simulated tropospheric $HNO_3$ originates from reactions with $NO_2$, a product of the photolysis of PAN. The difference in this diagonal structure between GLORIA and EMAC in the second part of the flight may result from a spatial displacement of the whole structure in the model, which is, however, unlikely due to the agreement of this $HNO_3$ structure in the first part of the flight,

and due to the agreement of structures in pollution trace gases in the second part of the flight (see below). It is more likely that EMAC overestimates the production of or underestimates the loss of $HNO_3$ here at altitudes below 14 km.

Comparisons between GLORIA $O_3$ and CAMS (Fig. 5d-e) show a general agreement between the two cross sections. In contrast to $HNO_3$, CAMS has reasonable stratospheric values for $O_3$ due to a satellite data assimilation scheme, which is only used for $O_3$ (Inness et al., 2015). The general distribution of $O_3$ is also well reproduced by the EMAC model, which does not assimilate satellite data, but applies a stratospheric chemistry scheme. Both models have difficulties in simulating the measured local enhancement at 5:30 UTC (purple box).

PAN is simulated similarly by CAMS and EMAC, and thus both models appear to have similar achievements and problems in reproducing the GLORIA measurements (Fig. 5g-i). The first part of the flight shows a local maximum (red box) above a local minimum (magenta box), neither of which are simulated. Both models instead simulate a, more or less, constant decrease in PAN with altitude. In the second part of the flight, both models succeed in reproducing the location of relative maxima (orange and cyan boxes) and of the minimum (blue box), but not the absolute VMRs. While GLORIA measured more than 500 pptv of PAN in the orange box, CAMS simulates 400 pptv and EMAC not more than 350 pptv. The background values of tropospheric PAN (blue box) are instead simulated higher (150 pptv) in CAMS and EMAC than measured (100 pptv). The measured enhancement above the tropopause (cyan box) is only visible in the models towards the end of the flight at approx. 1 km lower altitudes, and both models have lower maximum values (300 pptv for CAMS and 200 pptv for EMAC vs. 350 pptv for GLORIA).

$C_2H_2$ is only simulated by EMAC, which shows VMRs below 75 pptv, considerably lower than the measured VMRs (Fig. 5j-k). In the first part of the flight, EMAC simulates enhancements of up to 50 pptv below 14 km, where also GLORIA measurements show a local maximum of up to 150 pptv $C_2H_2$. In the second part of the flight, again a very small enhancement at the tropopause at 6:00 UTC (cyan box) is visible in EMAC, which is at the same geolocation as the enhancement in the measurements, but much weaker and less extensive. Measured maxima of $C_2H_2$ are not reproduced by the EMAC model.

For HCOOH (Fig. 5l-n), CAMS only simulates VMRs below 50 pptv, which are measured as background values by GLORIA. The spatial distribution in CAMS shows a tiny enhancement of up to 50 pptv close to the orange box, where GLORIA measured maximum values of more than 200 pptv. EMAC simulates HCOOH VMRs up to 125 pptv below 15 km altitude in the first part of the flight, where GLORIA measurements show no enhancements. The measured enhancement in the red box is not reproduced by the model, but a local maximum is simulated at the same altitudes later during this flight (at 4:30 UTC). In the second part of the flight, the measured maximum (orange box) is simulated as a local maximum, but with considerably lower VMRs of less than 75 pptv. In the averaged GLORIA cross sections, a small local maximum of 60 pptv is visible after 6:00 UTC, which coincides with the small enhancement in EMAC.

These comparisons of GLORIA measurements with CAMS and EMAC show the limitations of atmospheric chemistry model simulations. In the first part of the flight, for all gases other than $HNO_3$ in EMAC, not even the structure was correctly simulated. In the second part of the flight, models typically succeed in reproducing measured structures in the trace gases but with considerably lower absolute peak VMRs and higher background VMRs. As shown in Fig. 3, both parts of the flight were under similar convective influence. Both models, CAMS and EMAC, and their driving meteorological fields are not expected

to resolve all events of deep convection and therefore use parameterizations for convection. It may be possible that convective events responsible for pollution trace gas structures in the second part of the flight are better met by the model's meteorological fields and convection parameterizations than in the first part. For parameterizations of convection in EMAC, a large difference between different methods is reported for the upper troposphere (see Fig. 5, Tost et al., 2006). Other possible explanations of

the observed discrepancies during the different flight parts are the source regions and emission strengths that are considered in the models. It is possible that source regions responsible for measured enhancements of pollutants in the second part of the flight are considered by the models, while the regions responsible for measured enhancements of pollutants in the first part of the flight are not.

    As a simple sensitivity test, we have increased NMVOC emissions in the EMAC model globally by 50%. This was motivated

by Monks et al. (2018), who suggest that NMVOC emissions are globally underestimated. As expected, the new simulation results in Supplementary Fig. 20 have larger maximum VMRs of PAN, $C_2H_2$, and HCOOH, which in some cases match better to the GLORIA measurements. However, the overall structure in the trace gas distributions remains unchanged and still does not agree significantly better with the observations. In addition, the background VMRs of PAN and HCOOH are considerably overestimated in the sensitivity run. This simple test indicates that other uncertainties in the model are more

important for the mismatch to the GLORIA observations. These uncertainties in the models include errors in the concentrations of precursor species, uncertainties in or lack of implemented chemical reactions, temporal and spatial variability of the NMVOC emissions that are not considered by the emission inventories, and lack of or uncertainties in microphysical processes, such as scavenging. In addition, analyses of backward trajectories (see Supplementary Fig. 19 and Bucci et al., 2020) showed significantly different trajectory paths for ERA-Interim (used for the EMAC simulations) and ERA5. As shown by Bucci et al.

(2020), ERA5 explains pollution features better than does ERA-Interim. Based on these uncertainties, further model sensitivity studies are recommended to improve the agreement with the measurement, but are outside the scope of this study.

## 6   Conclusions

This study discusses the first simultaneous airborne measurements of $HNO_3$, $O_3$, PAN, $C_2H_2$, and HCOOH in high spatial resolution within the center of the AMA UTLS. The observations reveal air masses with strongly enhanced mixing ratios of

these pollutants with maximum VMRs of more than 500 pptv for PAN, and more than 200 pptv of $C_2H_2$ and HCOOH. In particular, a layer of enhanced PAN ($\approx$400 pptv) and $C_2H_2$ ($\approx$100 pptv) has been measured at and above the tropopause. From the atmospheric lifetimes and the trajectory analysis, it is estimated that these enhancements exist in the atmosphere for more than a few days. Other air masses below 15 km in which strongly enhanced concentrations of pollution trace gases have been observed are linked to recent convective events. These measurements and their analysis confirm that PAN, a precursor of $O_3$, is

efficiently transported upwards by convection, and transported for a longer time in the tropopause region, as shown earlier by Glatthor et al. (2007), Fadnavis et al. (2015), and Ungermann et al. (2016). In contrast to the study by Gottschaldt et al. (2018), no enhancements of $O_3$ are visible in the GLORIA measurements. However, the enhancements discussed by Gottschaldt et al. (2018) are within the estimated observational uncertainty and, thus, cannot be ruled out.

We found indications that HCOOH was washed out, while in the same air masses PAN, $C_2H_2$, and also water soluble $NH_3$ were transported to the measurement geolocation. By Höpfner et al. (2019), it is reported that a large amount of up to 1 ppbv of $NH_3$ was measured in the same air masses as the HCOOH minimum. Because IASI satellite total column measurements show maxima for both HCOOH and $NH_3$ at the estimated area of origin of these air masses (according to Höpfner et al., 2019), it is suggested that the wash out process in these air masses was very different for the two species, due to their different Henry's Law Constants and the pH of the droplets. However, other processes resulting in different UT transport characteristics of HCOOH and $NH_3$ have been discussed by Ge et al. (2018). Later during the flight, for air masses of different origins and histories, it is shown that enhanced HCOOH VMRs up to 200 pptv can reach altitudes up to 15 km without being washed out.

The analysis of backward trajectories from ATLAS and TRACZILLA shows that the two methods for the treatment of convection of these models highlight similar regions of enhanced convective upward transport. Backward trajectories, starting at the geolocations of measurements with local maxima and minima of NMVOCs, indicate various regions of enhanced convection and thus of the origin of these air masses. Air masses with enhanced PAN, $C_2H_2$, and HCOOH at altitudes between 12 and 15 km in the second part of the flight are connected with convection over India, while unpolluted air masses at similar altitudes are connected with convection events over sea and coastal areas. A comparison with measured OMI $NO_2$ enhancements indicates that some of these regions are plausible pollution sources. Similarly to Bucci et al. (2020), we find a strong convective influence of the transport of polluted air masses to the upper troposphere.

Comparisons of the measurements to CAMS and EMAC simulation results show that the first part of the flight appears to be under high convective influence, which the models do not account for. In the second part of the flight, CAMS and EMAC reproduce large scale structures of the spatial distributions of the measured trace gases, while the simulated peak VMRs of PAN, $C_2H_2$, and HCOOH are considerably lower compared to the measurements. Tropospheric background VMRs instead are simulated too high for PAN and HCOOH (the latter only for EMAC). An EMAC sensitivity test with NMVOC emission globally increased by 50%, did not considerably improve the agreement between simulation and measurements. Therefore, other uncertainties in the model apparently play a greater role: The emissions used in the model might not reproduce the temporal and spatial variability, convection events are not covered by the meteorological fields or model parameterizations, and chemical processes in the model might be uncertain. In addition, the T106 horizontal resolution still is rather coarse for these highly resolved measurements by GLORIA.

In our paper, we show that there are very fine-scale structures and a large variability of pollutant trace gases over horizontal scales of 200 km in the Asian monsoon UTLS. Some pollutants have been transported into the upper troposphere by convection within days before the measurements, while one part of the observed air masses remained at UTLS altitudes for a longer time. Atmospheric models have difficulties in reproducing the structures of the observed pollutant trace gas concentrations, likely because of uncertainties in the prescribed NMVOC emissions, rather coarse model resolution, and insufficient vertical transport from convection in the meteorological fields used to drive the model. Advanced schemes for convection detection along backward trajectories allow for the identification of source regions of the polluted air masses measured by GLORIA.

*Data availability.* GLORIA measurements are available in the database HALO-DB (https://halo-db.pa.op.dlr.de/mission/101) and on the KITopen repository (https://doi.org/10.5445/IR/1000125284). The CAMS model data is available from ECMWF (https://apps.ecmwf.int/data-catalogues/cams-reanalysis). OMI NO$_2$ level 3 data is available from NASA (https://doi.org/10.5067/Aura/OMI/DATA3007 Krotkov, 2013). The EMAC, ATLAS, and TRACZILLA data are available upon request.

5   *Author contributions.* SJ initiated the study, performed the analyses, and wrote the manuscript. MH, JU, GW, NG, and SJ performed the GLORIA data processing. FFV and EK operated GLORIA during the StratoClim campaign in Kathmandu. OK performed the EMAC simulations and designed the sensitivity study. SB and BL performed the TRACZILLA trajectory simulations. IW performed the ATLAS trajectory simulations. All authors commented on and improved the manuscript.

*Competing interests.* The authors declare that they have no conflict of interest.

10   *Acknowledgements.* We gratefully thank the StratoClim coordination team, in particular Fred Stroh, and Myasishchev Design Bureau for successfully conducting the field campaign. The results are based on the efforts of all members of the GLORIA team, including the technology institutes ZEA-1 and ZEA-2 at Forschungszentrum Jülich and the Institute for Data Processing and Electronics at the Karlsruhe Institute of Technology. We thank ECMWF for providing CAMS data and NASA for providing OMI data. The EMAC simulations were performed on the supercomputer ForHLR funded by the Ministry of Science, Research and the Arts Baden-Württemberg and by the Federal Ministry of 15   Education and Research. We thank Michelle Santee and one anonymous referee for improving this manuscript during the review process. We gratefully acknowledge support by the translation services at Karlsruhe Institute of Technology.

    Sören Johansson has received funding from the European Community's Seventh Framework Programme (FP7/2007–2013) under grant agreement 603557. We acknowledge support by the Deutsche Forschungsgemeinschaft and the Open Access Publishing Fund of the Karlsruhe Institute of Technology.

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
