# Peer review of "Pollution trace gas distributions and their transport in the Asian monsoon upper troposphere and lowermost stratosphere during the StratoClim campaign 2017"

_Atmospheric Chemistry and Physics, 2020_

## Referee Comment (RC1) · Michelle Santee (Referee) · 22 Jul 2020

**Review of "Pollution trace gas distributions and their transport in the Asian monsoon upper troposphere and lowermost stratosphere during the StratoClim campaign 2017"**
**by Johansson et al.**

This manuscript presents observations obtained by the GLORIA airborne imaging infrared limb sounder in the UTLS during the StratoClim field campaign, which investigated the 2017 Asian summer monsoon. Measurements of $HNO_3$, $O_3$, PAN, $C_2H_2$, and HCOOH are analyzed in detail. Two sets of back trajectory calculations using different models, each employing a novel scheme for detection of convective events, are used to identify source regions for the sampled air masses. GLORIA data are also used to evaluate the CAMS reanalysis and simulations from the EMAC chemistry climate model.

Overall, this is an interesting and valuable paper reporting on measurements from an important campaign, and I think it will be of interest to the broad community. Unfortunately, however, the manuscript is marred by many instances of unclear and awkward wording. This is not just a matter of style – the writing is confusing enough in places that the meaning the authors are trying to convey is obscured. Thus, in my opinion, the manuscript requires a substantial amount of "cleaning up" before it can be published, and I have compiled a rather large number of comments. In most cases my concerns can be allayed by simply correcting and clarifying the discussion, with few requiring additional analysis or other significant changes. In some places I have tried to offer suggestions to improve the clarity and readability of the text. But, although each point is perhaps minor when considered in isolation, in aggregate they add up to major revisions. In addition, although some minor wording and grammar corrections have been suggested below, the manuscript should be copy-edited to improve the English. In particular, many errors in the use of commas are present throughout the manuscript.

**Specific comments and questions: both major substantive issues and minor points of clarification, wording suggestions, and grammar / typo corrections are listed together for each Section in sequential order through the manuscript**

**Abstract**
- P1, L3-4: with base in --> based in
- P1, L7: This sentence does not make sense: why is the word "instead" used? Longer than what? No timescales have yet been mentioned.
- P1, L8: This line is misleading, since $NH_3$ is not presented in this paper. I suggest adding "previously reported" in front of "maximum".
- P1, L9: transport to the measured pollution trace gas occurrences --> transport on the measured pollution trace gases
- P1, L13: OMI should be spelled out here.
- P1, L16-17: It is not clear to me what the assertion that the models reproduce the large-scale structures of the pollutant distributions "if the convective influence on the measured air masses is captured by the meteorological fields used by these simulations" is based on, since this study does nothing to demonstrate that the models capture convective influence well,

and in fact numerous prior studies have shown that they do not. Perhaps the large-scale trace gas distributions are controlled mainly by the large-scale circulation, which global models do simulate reasonably well.

- P1, L17: Both models do not have --> Neither model has
- P1, L20: to reproduce --> in reproducing

**Introduction**

- P2, L4: The manuscript by Basha et al. (2019) has been rejected and should not be cited.
- P2, L6: The Santee et al. (2017) citation in the Reference list is an abstract for a symposium presentation (on a topic unrelated to this work) and is clearly not the intended reference, which should be a 2017 paper on the ASM published in JGR-Atmospheres.
- P2, L6-7: I don't think it is quite fair to characterize the vertical and horizontal resolution and sampling of satellite limb sounders as "low". Their sampling is vastly better than that of ground-based or airborne sensors, and their vertical resolution is much better than that of nadir sounders. Also, "low" is not the most suitable qualifier for "sampling". I suggest "relatively coarse" instead.
- P2, L7-8: Some references for the sentence about airborne in situ measurements inside the AMA would be appropriate.
- P2, L10-23: This paragraph as a whole is rather disjoint, with multiple independent thoughts assembled together with no thread connecting them. The last two sentences in particular seem out of place and do not follow from previous lines, and it's not clear why the last one begins with "However". I suggest rewriting to improve the cohesion and flow.
- P2, L14-15: Numerous papers have touched on this topic between Singh (1987) and Höpfner et al. (2019), so "e.g." is needed here.
- P2, L24-25: This sentence is overly general – it should be made more clear that it specifically refers to the region of the ASM, not the entire upper troposphere.
- P2, L28: "these data" could be interpreted as referring to the entire StratoClim dataset from all instruments, so: these data --> the data reported here
- P2, L32: are --> have been

**Section 2.1**

- P3, L20: The Santee et al. (1998) paper, which focuses on PSCs, is not really the best reference for MLS observations of HNO3 in the UTLS. A more relevant paper to cite for this point would be Santee et al. (JGR-Atmospheres, 2011).
- P3, L24-25: as stratospheric --> as a stratospheric; "within" is not the right word in this context; PAN is not defined until P4
- P3, L30: This is a very abrupt transition to tropospheric ozone; it would be better to say something about background values of tropospheric ozone, and possibly its sources as well, before talking about the magnitude of enhancements.
- P4, L4-5: Numerous papers (some of which are referenced elsewhere in this manuscript) have discussed the low abundances of ozone inside the AMA, so it is not appropriate to cite only a single paper for this point; at the very least an "e.g." is needed here.

- P4, L10-11: This sentence is somewhat inaccurate. Ozone is typically low inside the AMA; the Park et al. papers cited here use low ozone abundances (along with enhanced CO) as a marker of tropospheric air trapped inside the AMA. Park et al. (and others) have used larger abundances of ozone as an indicator of the presence stratospheric air, but not "polluted air" as stated here. If the authors are referring to the findings of Gottschaldt et al. (2017), then that paper should be cited here. In addition: measurements of $O_3$ … is --> $O_3$ is.
- P4, section 2.1.3: Typical background abundances of PAN should be stated here, as they are in the respective subsections for $HNO_3$ and $O_3$. This information is given in Section 3, but for completeness it should appear here as well.
- P4, L17-18: It is stated that photolysis plays a minor role, but according to Fadnavis et al. (ACP, 2015), photolysis is the dominant loss process for PAN in the UTLS. In addition, 250 K is not a higher altitude than 298 K.
- P4, L21: The CRISTA acronym needs to be defined on first use.
- P4, section 2.1.4: It is even more critical to help readers by providing some idea of typical background values for acetylene since that information is not given in Section 3.
- P4, L27: This sentence is awkward. I suggest reordering as: "Acetylene or ethyne ($C_2H_2$), a product of biofuel and fossil fuel combustion and biomass burning, has maximum tropospheric mixing ratios of a few pptv."
- P4, L30: The ATMOS acronym needs to be defined on first use.
- P5, L5: estimated to --> estimated to be
- P5, L8: or --> and
- P5, L9: measurements --> measurements of

**Section 2.2**
- P5, L11: performed with basis in --> conducted from a base in
- P5, L13-14: Why is only a single research flight singled out for analysis in this study? Unless some explanation is given about why the data available from the other three flights are not considered, readers may draw their own inferences about their quality or consistency.
- P6, Table 1 caption: Used spectral regions --> Spectral regions used
- P6, L8: I don't think that this sentence is completely clear. I suggest instead: "The retrieval strategy used here differs from that of Johansson et al. mainly in the applied …"
- P6, L12: substituted by retrieving --> replaced by retrieval of

**Section 2.3**
- P7, L4: atmosphere --> atmospheric; interaction --> interactions
- P7, L9: The grid specification should be written in a manner consistent with that in L29.
- P7, L10-15: These sentences are poorly written and unclear. Was the extension of the MECCA model performed by the authors as part of this work, or by the EMAC team? Are the values quoted for the number of reactions, etc., for the "standard" MECCA submodel or the "extended" one? It would be clearer to say "two sensitivity simulations with emissions of NMVOC increased by 50% and 100%". Other minor wording suggestions: in contrast to --> beyond that of; with regard to a better --> to improve the; photolyses --> photolysis reactions. ECMWF should be defined here, not in the following paragraph.

- P7, section 2.3.1: How are emissions prescribed in the EMAC runs done for this study? This information seems just as critical to me as the details of the chemical submodel. In particular, if emissions were prescribed using RCP scenarios, which do not include specific events, such as major fires in any given year, then even specified-dynamics EMAC simulations cannot be expected to replicate the observations closely.
- P7, section 2.3.2: Similarly, information about the emissions in CAMS also needs to be given.
- P7, L23-24: This sentence mentions a study evaluating the CAMS chemical reanalysis using aircraft measurements but provides no information about the results of those comparisons. Did Wang et al. (2020) find that CAMS fields match the measured species well or not? What are the implications for this work? In addition, the paper by Wang et al. has now been published, so the reference needs to be updated.
- P8, L9-10: ECMWF has previously been defined. Is this 3-h ERA5 product different from the one mentioned on P7, L28 with 1-h temporal resolution?
- P8, L16: The paper by Wohltmann et al. (2019) has now been published, so the reference should be updated.
- P8, L22-26: The investigation described in these sentences is interesting, but the results reported here are vague and their implications for this study are unclear (and the last sentence in this paragraph could also be better composed). What exactly is meant by "major differences" and "minor influences"? This discussion should be more quantitative. Do the findings from these ATLAS and TRACZILLA tests have any implications for the results from EMAC, since those runs were driven with ERA-I?
- P8, section 2.3.5: It is not appropriate to include the discussion of OMI tropospheric column $NO_2$ as part of Section 2.3, which is entitled "Atmospheric model simulation". Perhaps it should be in its own subsection. Alternatively, perhaps it could go in Section 2.1, "Measured trace gases". That section contains a general description of the species measured by GLORIA and analyzed in this study, but it could be slightly restructured to include the OMI $NO_2$ data.
- P8, section 2.3.5: It is necessary to provide information on the quality and resolution of the OMI tropospheric column $NO_2$ data, as well as a suitable reference for this specific product (beyond the general OMI instrument paper and the Krotkov (2013) citation, which is just for the L3 files and which is also incomplete).
- P8, L28: delete "instrument"
- P8, L30: troposheric --> tropospheric

**Section 3**
- P9, L4: Actually, $HNO_3$ strongly increases a few km above the tropopause, starting at about 19 km.
- P9, L6-7: This wording is unclear. By "local enhancements up to 0.5 ppbv", do the authors mean that the measured mixing ratios approach 0.5 ppbv, or that they are 0.5 ppbv larger than the regional background values (it looks like the latter to me). Some of these enhancements appear to be located at altitudes higher than 16 km. In fact, the particular structure noted at 4:00 UTC is at more like 16.5 km.
- P9, L8: Why is the magenta box drawn so as to exclude the peak in this enhancement at 4:00 UTC, and also the higher values right at 16 km just before 4:15 UTC? If this enhanced

structure is of interest for further analysis, I would think that it would be desirable to encompass the region of its strongest signature.

- P9, L14-15: It would be helpful if the colored boxes on Figure 2 were also overlaid on Supplementary Figures 2, 4, 6, 8, and 10.
- P9, L18: A local --> A PAN local
- P9, L32: with VMRs --> with $C_2H_2$ VMRs
- P11, L1: of the --> on the
- P11, L3-4: as for all other gases than $HNO_3$, considerably large HCOOH of more than 200 pptv is --> as for all gases other than $HNO_3$ and $O_3$, considerably larger abundances of HCOOH of more than 200 pptv are
- P11, L5-9: I'm wondering why the authors have chosen not to highlight the region with the minimum in HCOOH where PAN and $C_2H_2$ are present in its own colored box. Considerable discussion is devoted to this part of the flight, possibly more than for some of the regions that are enclosed within boxes.
- P11, L10-11: This presence of PAN and $C_2H_2$ and the absence of HCOOH --> The presence of PAN and $C_2H_2$ together with the absence of HCOOH
- P11, L14-15: The period after "liquid" should be moved to after "2016)".
- P11, L19-20: The authors need to clarify that they are not talking about "all discussed gases" in these lines, as stated, but only the tropospheric tracers.
- P11, L27: which is --> as
- P11, L28: of the --> on the; but not in HCOOH suggests --> but not in HCOOH, suggest

**Figure 2:**
- It would be extremely helpful to the reader to: (1) enlarge the major tick marks on both x- and y-axes, (2) add minor tick marks, and (3) include tick marks on the right-hand y-axis and the top x-axis. Without them, it is very difficult to judge the values quoted in the text.
- The colored boxes on both the maps and the curtain plots are a little hard to see, as is the green line marking the tropopause. Perhaps it would help to make these lines a bit thicker.
- Caption: Specify that the green line is on the cross section plots. Delete ", which are".

**Section 4**
- P11, L33: estimate --> identify; high --> strong
- P12, L4: aides --> aids
- P12, L11: Although the overlaid boxes in Figure 3 facilitate comparison with Figure 2, the authors should consider adding an altitude scale on the right-hand y-axis of the panels as well. It would also be helpful to state the approximate pressure level corresponding to 15 km in this line.
- P12, L16: Along these trajectories, regions are bordered orange, where the density of convective events along these trajectories --> Regions are outlined in orange where the density of convective events along these trajectories
- P12, L19: I. e. the smallest bordered regions --> That is, the smallest outlined regions; 1.0% and 10.0% --> 1.0% or 10.0%

- P12, L16-19: My apologies, but I am missing something here. I don't quite understand how the densities of convective events discussed in this paragraph relate to the convection probabilities shown in Figure 3 and discussed in the previous paragraph (which are an order of magnitude larger). Please clarify the relationship between these two quantities.
- P12, L20: I'm confused here too – why would it necessarily be the case that "larger regions contain accordingly a larger fraction"? A large region encompassed by a single colored contour but no inner contours would still have convective densities between 0.1% and 1.0%, no matter its size. Unless an inner contour is present, the fraction does not reach 1.0%. In addition, I have looked closely at Figure 4, and I am not convinced that any of the outlined regions contain the innermost contour representing 10%, except for one orange region in the TRACZILLA panel. Perhaps the rarity of that occurrence should be pointed out.
- P14, L1: as average over 14 days --> as an average over the 14 days
- P14, L6: delete "from the measurements"
- P14, L7: since $HNO_3$ is not a pollutant, it would be best to delete "otherwise".
- P14, L8-9: The flow in this paragraph needs to be improved. The sentence about the small fraction of trajectories experiencing convection in the 5 days leading up to the measurement is ambiguous; it immediately follows a sentence on the magenta region and thus appears to be about that area, but in fact I think it is referring to the red region. This should be clarified.
- P14, L10-12: The writing in these lines is very unclear. Assuming that I have interpreted them correctly, I suggest instead: "For most regions marked red, only the 0.1% contours are present; thus convective influence along the trajectories was weak. However, most regions marked red in northeastern China lie close to areas with enhanced $NO_2$, so these regions may possibly have contributed to the measured enhanced pollution trace gases."
- P14, L13-14: Again, I am confused about how the 30% value quoted here for the red regions can be reconciled with the 1% contour outlining those regions in Figure 4. The sentence in these lines is quite unclear. I'm also confused about exactly what is being shown in Supplementary Figure 12. As I understand it, the trajectories are launched from the GLORIA measurement locations, which in many/most cases are not characterized by ongoing convection. However, although the caption to Figure S12 is unclear, particularly the description of panel (c), it seems to suggest that a convective event was occurring at the time the trajectories were launched, and that 30% of those back trajectories had experienced convection leading up to that point. Please clarify.
- P14, L15-16: The magenta box is not shown on Fig. 2i, j, nor was a minimum in HCOOH in this region discussed (P11, L1-17). If anything, HCOOH looks slightly high in that area. I assume that "close to the red maximum" is referring to the pollutant enhancements in the red box?
- P14, L16-17: These two sentences are poorly written, but if I understand their meaning correctly, then I think that it would be much clearer to say: "Both trajectory models show similar convective densities as for the red regions above China, and they also show substantial convective activity above the South China and Philippine Seas. TRACZILLA also indicates regions of strong convection northwest of the flight path."
- P14, L17-18: regions only the 0,1% --> regions, only the 0.1%; influence of --> influence on; also, eliminate one of the instances of "again" in this sentence

- P14, L19-20: This sentence is badly written and hard to read. I suggest instead: "However, in this case, it is likely that convection in the regions above the South China and Philippine Seas brought up clean maritime air." But perhaps I have not understood this sentence. I can see that convective transport of clean maritime air could produce a local minimum in the pollutants, but how could it have led to enhanced $HNO_3$ in this region?
- P14, L23: coast, even the 1%, and for TRACZILLA also the 10% lines --> coast, the 1%, and for TRACZILLA even the 10%, convective density lines
- P14, L24-25: orange regions --> regions marked orange; Southern Chinese --> South China
- P14, L27: 50% convective --> 50% of convective
- P14, L28-29: This sentence is unclear. More plausible than what? More likely than what?
- P14, L30: Since the "previous one" discussed was a maximum, not a minimum, it would be better to say "as the region outlined in orange".
- P14, L31-32: indicates regions between the flight path and the Bay of Bengal as source region --> indicate convective source regions between the flight path and the Bay of Bengal; Southern Chinese and Philippine Sea --> South China and Philippine Seas
- P14, L32: low $NO_2$ measurements --> low $NO_2$
- P15, L1: delete the comma after "flight" and "in the measurements".
- P15, L2: According to --> Based on; similar to --> similar to that of the
- P15, L3-4: Why would bringing up relatively pristine marine boundary layer air lead to a local enhancement in ozone? Also: west India and above the South Chinese and Philippine Sea --> eastern India and above the South China and Philippine Seas.
- P15, L6-7: I do not follow the logic here. The relevant sentence in Section 3 "suggests that these air masses are older than a few days (lifetime of HCOOH), but younger than 2 weeks (lifetime of $C_2H_2$)". How does that lead to the statement here that "convection 10 days before the measurement only had a minor influence" – that is, where does the value of 10 days come from? Perhaps the authors mean "convection any time in the last two weeks"? Also: influence to --> influence on.
- P15, L8-10: This statement is slightly inaccurate, so it would be better to be more precise with the language here: "… Fig. 3, which does not show strongly enhanced convection probabilities in the cross sections for either trajectory model. ATLAS and TRACZILLA both only show a small convective region over the Philippine Sea for the cyan region of interest, and ATLAS also shows convective activity above central China."
- P15, L11: less than 20% convective events of all trajectories occurred at all --> less than 20% of all trajectories experienced any convective events
- P15, L15: I'm not sure what the take-away message for the reader is. Does the fact that both models seem to identify source regions that are less "plausible" call into question the entire source attribution analysis? Are these regions really less plausible as source regions because they are characterized by low OMI tropospheric column $NO_2$? As mentioned in connection with Section 2.3.5, some discussion of the reliability and sensitivity of these OMI data is needed. Moreover, can it necessarily be assumed that tropospheric column $NO_2$ is a robust proxy that reflects *all* possible sources for these NMVOCs? In particular, according to Section 2.1.5, formic acid arises in part from biogenic emissions. Would those be captured in the $NO_2$ measurements? Some further discussion is warranted here.

**Figures 3 and 4 and Supplementary Figure 12**

- Figure 3 caption: pressures below --> pressures less than
- Figure 4 caption: origin of regions of interest --> origin of air masses in the regions of interest; for 10 days of which the temporal evolution fo the first 5 days are --> for 10 days, of which the temporal evolution for the first 5 days is
- Figure S12 caption: Fig. 6b --> Fig. 4b

**Section 5**

- P15, L25-26: The EMAC HNO3 mixing ratios at the tropopause look quite a bit smaller than 0.75 ppbv to me.  In addition, the writing in this sentence is very awkward; I suggest rewriting as: "… flight; they decrease to values of 0.75 ppbv at the tropopause.  Simulated maximum stratospheric values are not always as high as those measured, but they agree to within …."
- P15, L27-28: Delicate … repeated --> Fine-scale … reproduced
- P15, L29: are reproduced --> is reproduced
- P15, L31-P16, L2: The authors posit that the diagonal feature in the $HNO_3$ field simulated by EMAC may originate from reactions with $NO_2$, and the tone of the discussion seems to suggest that this may be a model artifact, especially in the latter portion of the flight.  But they have made no attempt, here or in the previous section, to account for the similar feature seen in the GLORIA measurements in the first half of the flight.  What is the explanation for the observed structure in $HNO_3$?
- P16, L1: reactions with $NO_2$, product of the photolysis of PAN. Too high values below the tropopause are also simulated towards --> reactions with $NO_2$, a product of the photolysis of PAN. Values that are too large are also simulated below the tropopause towards
- P16, L5: It would be appropriate to include a reference for the CAMS assimilation of $O_3$.
- P17, L1: to simulated --> in simulating
- P17, L5: which is not simulated --> neither of which are simulated
- P17, L6: to reproduce --> in reproducing
- P17, L11: To my eye, PAN values measured in the region outlined in cyan were not higher than 350 pptv, not 450 pptv as stated here.  If this is meant to be a general statement (not specifically about the enhancement above the tropopause towards the end of the flight), then this sentence needs to be rewritten.
- P17, L12: which is considerably below --> considerably lower than
- P17, L15: In addition to pointing out that the EMAC $C_2H_2$ enhancement is in the same geolocation as the measured enhancement, it would be good to note that the simulated enhancement is much weaker and less extensive than the measured enhancement; it would also be helpful to add "(cyan box)" here.
- P17, L12-15: A point that is missing from the $C_2H_2$ discussion is the fact that EMAC completely fails to simulate the maxima in the red and orange boxes and the minimum in the magenta box, even in a relative sense.
- P17, L22: below --> of less than
- P17, L24: Even for $HNO_3$, the structure was correctly simulated by only one of the models.

- P17, L25-26: The authors state that their results indicate that the meteorological fields used to prescribe transport in the simulations do not include processes relevant for the observed situation. I presume that they are referring to deep convection, which is not resolved by the reanalyses, but that should be clarified. I am wondering, however, why this would be a factor only for the first part of the flight (which the sentence in question is about). According to Figure 3, as well as much of the discussion over the preceding pages of the manuscript, the second half of the flight was influenced by convection up to 150 hPa to a similar degree.
- P17, L30: to reproduce --> in reproducing
- P17, L32: within --> in

**Section 6**
- P18, L57: The writing in these sentences is clumsy. Moreover, I'm afraid that I don't follow the logic of the arguments. First, as mentioned in an earlier comment, both portions of the flight are characterized by high convection probabilities up to 150 hPa, so for that reason alone it doesn't make sense to focus only on the second half. Second, the authors appear to be saying that *because* the first part of the flight is strongly influenced by convection, the simulated results would not be affected by increased emissions. But that seems backwards to me – in the absence of convection, the strength of the surface emissions would be of little consequence. This discussion needs to be clarified.
- P18, L8: have comparable horizontal resolutions between --> obtain comparable horizontal resolution for both
- P18, L11: the emission scenario "+100%" --> the scenario with emissions increased by 100% ("+100%")
- P18, L13: E.g., --> For example,
- P19, L6-8: I'm not sure that it is true that GLORIA did not observe the slight enhancement in HCOOH at 6:00 UTC and 16 km. There may be a faint hint of this structure in the data. Perhaps this feature should have been introduced earlier in the discussion, e.g., P17, L16-22.
- P19, L10-14: Of course, although the increased emissions led to larger maximum values of PAN that matched the observed peak abundances better, they did nothing to improve the structure of the simulated field. I do not think that this is an unanticipated result. I would have expected background abundances of these tropospheric tracers to rise along with peak abundances in the increased-emissions scenario. So I am slightly puzzled by the discussion in these lines, which focuses on the impact of vertical resolution on the modeled fields. Its placement in this paragraph seems to imply that the smoothing effect of the coarser resolution of EMAC, which blunts peak abundances and blurs or erases fine-scale features, is somehow responsible for the background values of PAN being too high in this sensitivity test. In fact, I think that the resolution issue is just as relevant for the baseline model run, in which background abundances were also overestimated, and it would be more appropriate to move the discussion about it to Section 5.
- P19, L11-12: Anyhow, the tropospheric background values are modeled too high in both, the "+50%" and "+100%", simulation --> However, the tropospheric background values are substantially overestimated in both the "+50%" and the "+100%" simulations
- P19, L13: resolutions --> resolution

- P19, L14-15: reproduce on average and therefore smooths the finer resolved image --> reproduce on average, thereby smoothing the fine-scale structure
- P19, L15-16: with 100% increased NMVOC emissions --> with NMVOC emissions increased by 100%
- P19, L16: The possibility that model/measurement discrepancies may be partly attributable to emission sources not represented in the inventory used in these EMAC runs is mentioned. As I noted in connection with Section 2.3.1, which emission inventories were used in these simulations is a critical piece of information that has been omitted from the manuscript.
- P19, L18: That the meteorological reanalyses do not resolve local deep convection is a well-known issue that is presented here as a finding of this study. In addition, another aspect (besides the reanalyses) that does not appear to have been considered by the authors is the convective parameterization being used for these EMAC simulations. The choice of which convective parameterization is used has been shown to have a substantial impact on modeled trace gas distributions.
- P19, L18: indicates convective events that are not resolved in the meteorological fields that are prescribing dynamics --> indicates the occurrence of convective events that are not resolved in the meteorological fields used to prescribe dynamics

**Figure 6**
- Caption: EMAC with 50% (middle column), and EMAC with 100% increased NMVOC emissions (right column) distributions --> EMAC distributions with NMVOC emissions increased by 50% (middle column) and 100% (right column)

**Conclusions**
- P19, L23: In my opinion, the statement that this study discusses "the first measurements of $HNO_3$, $O_3$, PAN, $C_2H_2$, and HCOOH in the center of the AMA UTLS" is too broad. While that may be true for some species of the species listed, it is not true for all of them. This statement should be qualified in some way, e.g.: first airborne measurements, or first measurements by GLORIA.
- P19, L27-28: below 15 km with strongly enhanced pollution trace gases measured are linked to recent convective events as transport mechanism --> below 15 km in which strongly enhanced pollution trace gases were measured are linked to recent convective events as the transport mechanism
- P19, L29-30: This study is not the first to show that PAN is efficiently transported to the UTLS by deep convection, as is implied by the wording in these lines.
- P19, L31: our measurements --> the GLORIA measurements (this is especially important for readers who may focus just on the Conclusions, since the instrument has not yet been identified in this section).
- P20, L2: was --> were
- P20, L3: at the same air masses of --> in the same air masses as
- P20, L4-5: show for both species, HCOOH and $NH_3$ maxima at --> show maxima for both HCOOH and $NH_3$ at; this air masses --> these air masses; these species --> the two species

- P20, L6-8: Some of the discussion here is appearing for the first time in this manuscript. I do not think that it is appropriate to introduce new concepts in a section entitled "Conclusions".
- P20, L16: indicate --> indicates
- P20, L23-25: As noted earlier, the fact that EMAC overestimates tropospheric background mixing ratios is not unique to the increased-emissions scenario – it was also the case for the baseline run, and increased emissions are expected to affect background as well as peak abundances. The same comment regarding vertical resolution applies here as well.
- P20, L29: course --> coarse
- P20, L30: delicate --> fine-scale
- P20, L31-34: These sentences are poorly written. "enhancements" are not transported upward – pollution is transported upward, leading to enhancements in the UTLS. Likewise, a "region" is not transported "around the tropopause" – the measured air masses in that region are transported. And I'm not sure what is meant by "around the tropopause". Also: the origin of the measured species, which is likely to be caused by uncertainties of the --> the origin of the measured enhancements, likely because of uncertainties in the
- P20, L35: meteorological fields --> the meteorological fields used to drive the model
- P21, L1: estimation of origin for --> identification of source regions of

---

## Referee Comment (RC2) · Anonymous Referee #2 · 30 Jul 2020

Johansson et al. presented atmospheric trace gas measurements of HNO3, O3, PAN, C2H2 and HCOOH from the GLORIA instrument collected during the recent Strato-Clim campaign. These are valuable measurements from the Asian Summer Monsoon region and should be published in time. I have a few major comments. Number one being the use of language in writing. I agree with the other reviewer. Please have the manuscript edited by a professional writer before submission of revision. The other two major comments are related to the discussion in Section 4, pages 14-15 and the model evaluation approach in Sections 5 & 6. See below for details. I found the model evalu-

ation and bias assessment being the most problematic weakness of this study. These concerns need to be addressed before the paper should be accepted for publication.

P2, L6 & L12: Santee, 2007 → Santee et al., 2007.

P2, L16-17, what's the name of the aircraft campaign?

P2, L22-24. The logical connections of these two sentences and the previous section seems to be amiss. What's the relationship between radiative heating rates and transport in reanalyses with trace gases? Observations are sparse, so what? How do observations help? The last sentence in the next paragraph (atmospheric chemistry models . . .) is out of place. It fits much better in this paragraph, instead.

P2, L24-28. These three sentences seem to repeat themselves in various ways. It can be easily condensed into a single sentence but capture all essential elements. Please revise.

P2, L33. How about change to ". . . we use the NMVOC measurements from GLORIA collected during StratoClim to address two important science objectives."?

P3, L1. Which two models? EMAC and CAMS? Please specify. And I am sure this is not the "first evaluation" of these two atmospheric chemistry models.

P3, L7-8, you can simply say "Section 5 presents . . . & Section 6 discusses ..."

P3, L11. You probably should list the names of the five targeting species here.

P3, L22: → (MIPAS, von Clarmann et al., 2009)

P3, L25. "within the transport . . ." → "to examine the transport . . ."

P3, L30. WMO, 2019 → This one is actually "WMO, 2018"

P4, L1. Catalytic reactions with nitrogen oxides is a sink of ozone in the stratosphere and to some extent in the upper troposphere. Primary sources of ozone is the troposphere is in situ production of NOx +HOx, and NOx + peroxide radicals from VOC

oxidation, hence indicator of polluted air masses.

P4, R1. To be more accurate, you should change "->" to "<->"

P4, P17. Change to "the lifetime of PAN is very short at lower altitudes due to rapid thermal decomposition"

P4, L27. Maximum tropospheric mixing ratios of a few ppt for C2H2? Are you sure it is not a few ppb?? Check Xiao et al., (2007).

P7, L11. With regard to a better → with a better

P7, L14. Can you describe what are the NMVOC emission sources used in EMAC? When you say 50% and 100% additional emissions, do you mean from all emission sources, e.g. biomass burning, biofuel, fossil fuel, etc., and globally or just over Asia?

P8, L3-4. I find this sentence very awkward, with no clear description of what was actually done.

P7-8. In sections 2.3, could you also provide the details on which year, time period of the model simulations that were conducted?

P8. Section 2.3.5 is out of place. This is observations and it should be listed in Section 2.1 or Section 2.2, not in the modeling subsection.

P9, L3 & Figure 2. I think it is more accurate to say these are colored boxes are "air masses" of interest, rather than "regions" of interest. Also in figure 2 caption, add "shows" after "the green line". I find the green line very hard to see. A thick solid dark gray line would be much better. It also distinguishes its functionality from the color boxes.

P9, L21. Discriminate -> distinguish

P9, L27. As for Pan –> Similar to PAN

P9, L28. Delete "as for PAN"

P9, L32. -> there is a minor local maximum with VMRs up to 100 pptv.

P10, Figure 2. I find all panels very noisy, which is not surprising due to the large errors in GLORIA measurements as listed in Table 1. I would suggest average the measurement samples to larger temporal and vertical bins. This way you can average down the noise and illustrate the discussed features much better. In the present form, these features are barely distinctive from the surrounding background air masses. This is particularly problematic from C2H2 and HCOOH.

P11, L12 and hereafter. Please use the proper terminology: Henry coefficient should be changed to Henry's Law Constant.

P11, L14-15. You used two "while" in one sentence, grammatically not correct.

P12, Figure 3 and the corresponding discussion. (a) In the text, the relevant discussion uses km as a unit while the y-axis only shows pressure. Please add the corresponding km on y-axis. (b) The cyan box in TRACZILLA show likely convective influences while ATLAS shows none. Why the two models are showing such different results? And how can the GLORIA measurements help in assessing which back trajectory model is more accurate. Also, overall, I can see TRACZILLA shows more convective influences that ATLAS. How can you assess which one is more accurate?

P14-P15, the discussion on various air mass signatures. For clarity and easy-to-follow purposes, I highly recommend you assemble all this information into a table. In the table, please list the type of targeting air masses, altitude at which they are sampled, surface regions where they were originated from, average measured HNO3, O3, PAN, C2H2, HCOOH concentrations within these colored boxed, transport time since they left the surface, etc. Second, please add a summary discussion on the different chemical signature of airmasses from different regions, e.g. the purple/blue box air from the marine background vs. the orange/red box air from China, etc.

Sections 5 & 6. I found the observation-model comparison and evaluation a major

weakness of this study. Neither CAMS nor EMAC produces well the observed features and gradients of all five species. This is particularly the problem for C2H2 and HCOOH. I also have problems with the brutal way of increasing NMVOC emissions by 50% and 100%. I don't see any improvements in model performance with such approach. By matching with observations better in a few patchy spots, you are also creating huge biases in other places (Figure 6) for all three species. PAN, C2H2 and HCOOH can be emitted and/or formed from various sources, i.e. anthropogenic emissions and biomass burning emissions being the highly relevant sources. The differences in the regional distributions of these sources can have a dominant impact on tropospheric distribution of these gases after they are being lofted and formed during transport. A proper way to address this model bias is to adjust the emission strength of these individual sources in separate runs and assess how do the resulted distribution change. This way, one can potentially assess the sources of these biases. You are only presenting analysis of one single flight. Therefore, such model sensitivity simulations can be easily conducted within a few days. The new model results and the corresponding discussion should be included in the revised manuscript before the paper is moving forward for publication.

---

## Author Comment (AC1) · 22 Sep 2020

We thank Michelle Santee for her thorough and very valuable comments and suggestions. We changed all minor language and wording corrections according to her suggestions without listing all of the changes in this answer. Instead, a latexdiff document that tracks all changes made in the revised manuscript is provided in the author's response file.

To our knowledge, Copernicus will have the manuscript copy-edited by a professional writer in case of acceptance and before publication in ACP. In addition, translation

services at our institution (KIT) checked the language of the revised manuscript.
Our answers are given below. The original referee comment is repeated in **bold**, changes in the manuscript text are printed in *italics*.

**P1, L16-17: It is not clear to me what the assertion that the models reproduce the large-scale structures of the pollutant distributions "if the convective influence on the measured air masses is captured by the meteorological fields used by these simulations" is based on, since this study does nothing to demonstrate that the models capture convective influence well, and in fact numerous prior studies have shown that they do not. Perhaps the large-scale trace gas distributions are controlled mainly by the large-scale circulation, which global models do simulate reasonably well.**
We agree with the referee that we have not unambiguously demonstrated the influence of convective events to be responsible for the disagreement between models and GLORIA observations. Given the changes in the main part of the manuscript, we changed this part of the abstract to: *It is shown that these simulation results are able to reproduce large scale structures of the pollution trace gas distributions for one part of the flight, while the other part of the flight reveals large discrepancies between models and measurement. These discrepancies possibly result from convective events that are not resolved or parameterized in the models, uncertainties in the emissions of source gases, and uncertainties in the rate constants of chemical reactions.*

**P2, L4: The manuscript by Basha et al. (2019) has been rejected and should not be cited.**
We thank the referee for pointing that out! We missed to check this reference before the submission of the manuscript.

**P2, L7-8: Some references for the sentence about airborne in situ measurements inside the AMA would be appropriate.**

We added the Bourtsoukidis et al., 2017 (10.5194/amt-10-5089-2017) and Gottschaldt et al., 2018 (10.5194/acp-18-5655-2018) references. Because both references (and other references we know) only describe measurements of air masses of the AMA edge outflow, we rephrased to: *Airborne in-situ observations of air masses belonging to the AMA are extremely sparse and often sample only filaments, border areas, or outflow of the AMA (e.g., Bourtsoukidis et al., 2017, Gottschaldt et al., 2018.)*

**P2, L10-23: This paragraph as a whole is rather disjoint, with multiple independent thoughts assembled together with no thread connecting them. The last two sentences in particular seem out of place and do not follow from previous lines, and it's not clear why the last one begins with "However". I suggest rewriting to improve the cohesion and flow.**
According to the comments of both referees, we reformulated and restructured this paragraph into two paragraphs: The second paragraph of the introduction now mentions studies about the transport in the ASM, focusing on open issues of vertical transport. The third paragraph of the introduction now summarizes studies of pollution trace gas measurements (and their implications) for the ASM UTLS.

**P3, L30: This is a very abrupt transition to tropospheric ozone; it would be better to say something about background values of tropospheric ozone, and possibly its sources as well, before talking about the magnitude of enhancements.**
We added typical $O_3$ VMRs in the ASM and restructured the paragraph, so that the sources are discussed before the magnitude of enhancements.

**P4, L4-5: Numerous papers (some of which are referenced elsewhere in this manuscript) have discussed the low abundances of ozone inside the AMA, so it is not appropriate to cite only a single paper for this point; at the very least an "e.g." is needed here.**

We added an "e.g.", and Santee et al., 2017 and Brunamonti et al., 2018 as additional references.

**P4, L10-11: This sentence is somewhat inaccurate. Ozone is typically low inside the AMA; the Park et al. papers cited here use low ozone abundances (along with enhanced CO) as a marker of tropospheric air trapped inside the AMA. Park et al. (and others) have used larger abundances of ozone as an indicator of the presence stratospheric air, but not "polluted air" as stated here. If the authors are referring to the findings of Gottschaldt et al. (2017), then that paper should be cited here. In addition: measurements of O3 . . . is –> O3 is.**
We rephrased this sentence. In addition, we added the suggested reference: *Similarly to $HNO_3$, enhanced $O_3$, within the generally $O_3$ poor AMA upper tropospheric air, is either interpreted as indicator of stratospheric air (e.g., Park et al.,2007,2008) or connected to uplift of $O_3$ precursor species of polluted air (Gottschaldt et al., 2017).*

**P4, section 2.1.3: Typical background abundances of PAN should be stated here, as they are in the respective subsections for HNO3 and O3. This information is given in Section 3, but for completeness it should appear here as well.**
According to the referee's suggestion we added: *Typical background abundances of PAN in the upper troposphere are below 100 pptv (Glatthor et al., 2007).*

**P4, L17-18: It is stated that photolysis plays a minor role, but according to Fadnavis et al. (ACP, 2015), photolysis is the dominant loss process for PAN in the UTLS. In addition, 250 K is not a higher altitude than 298 K.**
Our original statement was meant for the whole troposphere. We added Fadnavis et al. (ACP,2014) as reference for this sentence and formulated more precisely: *[...] and photolysis play a minor role for lower tropospheric altitudes ( e.g., Fadnavis et al., 2014). In the upper troposphere instead, photolysis is the dominant loss process for PAN (e.g., Fadnavis et al., 2015).*

In addition, we clarified, that the numbers given are temperatures, not potential temperatures.

**P4, section 2.1.4: It is even more critical to help readers by providing some idea of typical background values for acetylene since that information is not given in Section 3.**

We added: *Typical background values for $C_2H_2$ are below 75 pptv (e.g., Xiao et al., 2007; Wiegele et al., 2012).*

**P5, L13-14: Why is only a single research flight singled out for analysis in this study? Unless some explanation is given about why the data available from the other three flights are not considered, readers may draw their own inferences about their quality or consistency.**

We added these sentences for explanation: *This research flight was selected for this work due to high flight altitudes and low cloud top altitudes within the AMA, which are both optimal measurement conditions for the infrared limb instrument GLORIA. This research flight was by far the best, due to the flight length allowing different air masses to be sampled and the low cloud top altitude.*

**P7, L10-15: These sentences are poorly written and unclear. Was the extension of the MECCA model performed by the authors as part of this work, or by the EMAC team? Are the values quoted for the number of reactions, etc., for the "standard" MECCA submodel or the "extended" one?**

Sorry for the confusion, we selected a more comprehensive chemistry set-up as usual in our simulations. The MECCA submodel was not extended. We removed the word "standard", because we do not explain it in the text and changed the sentence accordingly: *The chemical setup of the chemistry submodel MECCA (Sander et al., 2011) was selected with focus on the simulation of PAN and tropospheric chemistry.*

**P7, section 2.3.1: How are emissions prescribed in the EMAC runs done for this study? This information seems just as critical to me as the details of the chemical submodel. In particular, if emissions were prescribed using RCP scenarios, which do not include specific events, such as major fires in any given year, then even specified-dynamics EMAC simulations cannot be expected to replicate the observations closely.**

The referee is right, the emissions do not include the specific events of the year 2017. We use an emission scenario, which is quite common in the climate modeling community and currently, we do not have more recent emission data for the year 2017. We will express this more clearly in the paper. Nevertheless, we are convinced that the EMAC results should remain in the paper, because we think that simulation results based on these commonly used emission scenarios should be compared to measurements.

**P7, section 2.3.2: Similarly, information about the emissions in CAMS also needs to be given.**

We added: *Anthropogenic emissions are prescribed by MACCity (MACC/CityZEN; Granier et al., 2011), biogenic emissions by MEGAN2.1 (Model of Emissions of Gases and Aerosols from Nature; Guenther et al., 2012), and biomass burning emissions by GFAS v1.2 (Global Fire Assimilation System; Kaiser et al., 2012).*

**P7, L23-24: This sentence mentions a study evaluating the CAMS chemical reanalysis using aircraft measurements but provides no information about the results of those comparisons. Did Wang et al. (2020) find that CAMS fields match the measured species well or not? What are the implications for this work? In addition, the paper by Wang et al. has now been published, so the reference needs to be updated.**

Thanks for reminding us of the updated Wang et al., 2020 paper. We added to the section: *Profiles of $O_3$, $HNO_3$, and PAN above Hawaii showed an agreement within the*

*uncertainties of measurement and model. These agreements encourage the model evaluation of this study at altitudes of the upper troposphere in the ASM.*

**P8, L9-10: Is this 3-h ERA5 product different from the one mentioned on P7, L28 with 1-h temporal resolution?**
Both trajectory models (TRACZILLA and ATLAS) use the same ERA5 product, but ATLAS used a 3 h temporal resolution and a different spatial grid. We changed the manuscript to: *Trajectories from the ATLAS model (Wohltmann et al., 2009) are driven by the same ECMWF ERA5 meteorological fields as TRACZILLA, but with a temporal resolution of 3 h and a horizontal resolution of 1.125° × 1.125°.*

**P8, L22-26: The investigation described in these sentences is interesting, but the results reported here are vague and their implications for this study are unclear (and the last sentence in this paragraph could also be better composed). What exactly is meant by "major differences" and "minor influences"? This discussion should be more quantitative. Do the findings from these ATLAS and TRACZILLA tests have any implications for the results from EMAC, since those runs were driven with ERA-I?**
We added an additional figure to the supplement (Suppl. Fig. 19) to exemplarily show the influence of the reanalyses, trajectory type, and diffusion on the trajectory paths and location of convective events. We now refer to this supplementary figure and rephrased these sentences. In addition, we now reference to this investigation in the discussion of possible improvements of the EMAC simulations: *In an analysis of the ATLAS trajectories, the influence of the usage of ERA5 or ERA-Interim as meteorological fields, the influence of applied vertical diffusion, and the influence of the usage of kinematic or diabatic trajectories was investigated (shown in the supplementary information). This analysis (and also similar analyses by Legras and Bucci (2019)) revealed that major differences occur between ATLAS trajectories that use ERA5 or ERA-Interim meteorological fields. These major differences are exemplarily visible*

*in Supplementary Fig. 19, where trajectory paths and locations of convective events are considerably different between ERA-Interim and ERA5. Compared to these large discrepancies, differences in trajectory paths and locations of convective events due to the usage of kinematic or diabatic trajectories, or due to the application of vertical diffusion are small.*

**P8, section 2.3.5: It is necessary to provide information on the quality and resolution of the OMI tropospheric column NO2 data, as well as a suitable reference for this specific product (beyond the general OMI instrument paper and the Krotkov (2013) citation, which is just for the L3 files and which is also incomplete).**

We added: *The version 3 standard retrieval of tropospheric column $NO_2$ comes with a spatial resolution of $1.0°$ × $1.25°$ (latitude × longitude), and showed an overall agreement with other satellite and ground based measurements of $NO_2$ (Krotkov et al., 2017).*

The Krotov (2013) citation was meant as a documentation of the data file we used for this work, which is encouraged to be used by ACP. Due to technical issues, the DOIs were not displayed in the bibliography, which is now fixed.

**P9, L6-7: This wording is unclear. By "local enhancements up to 0.5 ppbv", do the authors mean that the measured mixing ratios approach 0.5 ppbv, or that they are 0.5 ppbv larger than the regional background values (it looks like the latter to me). Some of these enhancements appear to be located at altitudes higher than 16 km. In fact, the particular structure noted at 4:00 UTC is at more like 16.5 km.**

We tried to formulate more precisely: *In the first part of the flight (until 4:45 UTC), also a local maximum of VMRs up to 1.0 ppbv is visible below the tropopause at altitudes between 15.5 km and 17 km (close to the red box in Fig. 2b).*

**P9, L8: Why is the magenta box drawn so as to exclude the peak in this enhancement at 4:00 UTC, and also the higher values right at 16 km just before 4:15 UTC? If this enhanced structure is of interest for further analysis, I would think that it would be desirable to encompass the region of its strongest signature.**

We added the (slightly adjusted) red box to the $HNO_3$ cross section plot and clarified:
*This maximum is continued by enhancements noted at 16 km at 4:00 UTC moving down to 15 km at 4:15-4:50 UTC with VMRs up to 0.75 ppbv (marked with a magenta box). The shape and position of the red and magenta boxes are optimized for the pollution trace gases PAN and $C_2H_2$ discussed later in this section to have a local maximum in the red and a local minimum in the magenta box. Thus, these boxes do not exactly match the structure in $HNO_3$. In addition, Höpfner et al. (2019) reported enhanced ammonium nitrate abundances in the red air masses, and a local minimum of ammonium nitrate in the magenta box. Given these different pollution trace gas and aerosol concentrations in the red and magenta boxes, it is assumed that these air masses have different origin, even though the structure in $HNO_3$ appears to be connected.*
The adjustment of the red box induced changes in Sec. 4.

**P9, L14-15: It would be helpful if the colored boxes on Figure 2 were also overlaid on Supplementary Figures 2, 4, 6, 8, and 10.**
We updated these Supplementary Figures according to the referee's suggestion. In addition, we refined the statement about the $O_3$ error within the purple box.

**P11, L5-9: I'm wondering why the authors have chosen not to highlight the region with the minimum in HCOOH where PAN and C2H2 are present in its own colored box. Considerable discussion is devoted to this part of the flight, possibly more than for some of the regions that are enclosed within boxes.**
We added a green box to highlight this minimum in HCOOH. We know, that the green

color might be difficult to see on top of the cross section, but with the white border all colored boxes have, it should be possible. We decided for this color because it is also easy to separate from the other colors in the written discussion later in the manuscript.

**Figure 2: It would be extremely helpful to the reader to: (1) enlarge the major tick marks on both x and y-axes, (2) add minor tick marks, and (3) include tick marks on the right-hand y-axis and the top x-axis. Without them, it is very difficult to judge the values quoted in the text**
We changed the figure (and similar figures later in the manuscript) according to the suggestions.

**The colored boxes on both the maps and the curtain plots are a little hard to see, as is the green line marking the tropopause. Perhaps it would help to make these lines a bit thicker.**
We increased line thicknesses according to the suggestions. In line with suggestions from the second referee, we also changed the color of the 380 K tropopause line to dark gray.

**P12, L11: Although the overlaid boxes in Figure 3 facilitate comparison with Figure 2, the authors should consider adding an altitude scale on the right-hand y-axis of the panels as well. It would also be helpful to state the approximate pressure level corresponding to 15 km in this line.**
We followed the suggestion of the referee and added an additional y-axis with an approximation of altitude to the plots. In addition, we also mentioned the corresponding pressures in the text.

**P12, L16-19: My apologies, but I am missing something here. I don't quite understand how the densities of convective events discussed in this paragraph relate to the convection probabilities shown in Figure 3 and discussed in the**

**previous paragraph (which are an order of magnitude larger). Please clarify the relationship between these two quantities.**

We have substantially rephrased the complete paragraph to make more clear what is shown in Figures 3 and 4 (also in response to your comment on P12, L20). The text was confusing and did not contain sufficient information for the reader to understand the method and the figures. In addition, there was a factual error in the text which increased the confusion: "the smallest bordered regions include at least 0.1% of convective events" should have been "the outermost contours include at least 0.1% (per square degree) of convective events" (i.e. just the opposite of what was written). We have also made more clear now that the unit of the fractions shown in Figure 4 is "percent per square degree", i.e. the quantity shown is a fraction per area and not just a fraction.

**P12, L20: I'm confused here too – why would it necessarily be the case that "larger regions contain accordingly a larger fraction"? A large region encompassed by a single colored contour but no inner contours would still have convective densities between 0.1% and 1.0%, no matter its size. Unless an inner contour is present, the fraction does not reach 1.0%. In addition, I have looked closely at Figure 4, and I am not convinced that any of the outlined regions contain the innermost contour representing 10%, except for one orange region in the TRACZILLA panel. Perhaps the rarity of that occurrence should be pointed out.**

The statement was incorrect and we have rephrased the paragraph (see also reply to P12, L16-19).

**P14, L8-9: The flow in this paragraph needs to be improved. The sentence about the small fraction of trajectories experiencing convection in the 5 days leading up to the measurement is ambiguous; it immediately follows a sentence on the magenta region and thus appears to be about that area, but in fact I think**

**it is referring to the red region. This should be clarified.**
We removed the reference to the magenta region, which is not needed in this paragraph. We apologize for the confusion.

**P14, L10-12: The writing in these lines is very unclear. Assuming that I have interpreted them correctly, I suggest instead: "For most regions marked red, only the 0.1% contours are present; thus convective influence along the trajectories was weak. However, most regions marked red in northeastern China lie close to areas with enhanced NO2, so these regions may possibly have contributed to the measured enhanced pollution trace gases."**
We changed the manuscript in line with the referee's suggestion. In addition, we changed the order of words in the first sentence to make cause and effect more clear: Because convective influence was weak, only the 0.1 percent per square degree contour is present.

**P14, L13-14: Again, I am confused about how the 30% value quoted here for the red regions can be reconciled with the 1% contour outlining those regions in Figure 4. The sentence in these lines is quite unclear. I'm also confused about exactly what is being shown in Supplementary Figure 12. As I understand it, the trajectories are launched from the GLORIA measurement locations, which in many/most cases are not characterized by ongoing convection. However, although the caption to Figure S12 is unclear, particularly the description of panel (c), it seems to suggest that a convective event was occurring at the time the trajectories were launched, and that 30% of those back trajectories had experienced convection leading up to that point. Please clarify.**
We have substantially rephrased the text and the caption in the supplement. In particular, we did not want to suggest that a convective event was occurring at the time the trajectories were launched, which is not the case. We changed the text in the manuscript to: *For the ATLAS model, it is shown in Supplementary Fig. 12 that for the*

[Figure]

*red region, less than 30% of all started trajectories experienced a convective event within 10 days before the measurement, showing the weak convective influence.*
In addition, we changed the caption of Fig. S12 to: *In b) and c), dots mark the location of all convective events experienced by backward trajectories starting in the red region (with the convection scheme switched on). b) is color-coded with the time difference between the convective event and the time of measurement, and c) is color-coded with the percentage of the other backward trajectories that already had experienced convection when the trajectory represented by the dot went into convection.*

**P14, L15-16: The magenta box is not shown on Fig. 2i, j, nor was a minimum in HCOOH in this region discussed (P11, L1-17). If anything, HCOOH looks slightly high in that area. I assume that "close to the red maximum" is referring to the pollutant enhancements in the red box?**
We thank the referee for pointing that out! HCOOH appears in that list by mistake. We removed it from this paragraph. We changed the formulation "close to the red maximum" to *close to the maximum of the pollutant species marked with the red box*.

**P14, L19-20: This sentence is badly written and hard to read. I suggest instead: "However, in this case, it is likely that convection in the regions above the South China and Philippine Seas brought up clean maritime air." But perhaps I have not understood this sentence. I can see that convective transport of clean maritime air could produce a local minimum in the pollutants, but how could it have led to enhanced HNO3 in this region?**
We changed the sentence according to the referee's suggestion. In addition, for the explanation of the enhanced $HNO_3$, we add: *Enhanced $HNO_3$ concentrations within these air masses possibly result from reaction of lightning $NO_x$ with OH to $HNO_3$ (see e.g., Schuhmann et al., 2007).*
We compared typical lifetimes of $NO_x$ in the upper troposphere (4-7 days according to Schumann et al., 2007; 10.5194/acp-7-3823-2007) with the time since the convective

event above the South China and Philippine Seas for the magenta air masses (3-5 days; see Suppl. Tab. 1). Together with observations of several ppbv of lightning $NO_x$ (Schuhmann et al., 2007), and $HNO_3$ as main sink of lightning $NO_x$, we consider this to be the most likely origin of the enhanced $HNO_3$ concentrations. In addition, we added in response to the referee's comment on "P9, L8" a comment on the structure of $HNO_3$ in the first part of the flight.

**P14, L28-29: This sentence is unclear. More plausible than what? More likely than what?**
We rephrased this sentence to: *This corresponds to the orange region in India with enhanced $NO_2$ columns.*
Other information in the original sentence was redundant to preceding sentences.

**P15, L3-4: Why would bringing up relatively pristine marine boundary layer air lead to a local enhancement in ozone?**
We added an interpretation of this result from the trajectory analysis: *These areas marked by the trajectories show low OMI $NO_2$ and indicate relatively clean boundary layer air, which cannot explain the measured local enhancement of $O_3$. This suggests that the measured local maximum of $O_3$ is of other than convective origin; possibly, the measured maximum is a pollution remainder transported for more than 10 days, or an intrusion of stratospheric air.*

**P15, L6-7: I do not follow the logic here. The relevant sentence in Section 3 "suggests that these air masses are older than a few days (lifetime of HCOOH), but younger than 2 weeks (lifetime of C2H2)". How does that lead to the statement here that "convection 10 days before the measurement only had a minor influence" – that is, where does the value of 10 days come from? Perhaps the authors mean "convection any time in the last two weeks"?**
This sentence was confusing and we changed it to: *In Sec. 3, it is suggested that*

[Figure]

*these air masses are transported for more than a few days, but for less than two weeks. For this reason, it is not expected to see strong convective influence in the trajectories a few days prior to the measurement.*

**P15, L15: I'm not sure what the take-away message for the reader is. Does the fact that both models seem to identify source regions that are less "plausible" call into question the entire source attribution analysis? Are these regions really less plausible as source regions because they are characterized by low OMI tropospheric column NO2? As mentioned in connection with Section 2.3.5, some discussion of the reliability and sensitivity of these OMI data is needed. Moreover, can it necessarily be assumed that tropospheric column NO2 is a robust proxy that reflects \*all\* possible sources for these NMVOCs? In particular, according to Section 2.1.5, formic acid arises in part from biogenic emissions. Would those be captured in the NO2 measurements? Some further discussion is warranted here.**
We rephrased and extended the last paragraph of this section, after a summary of air mass origins (as asked for by referee 2): *The comparison of ATLAS and TRACZILLA calculations of convective origin of the measured pollution species shows that there are few differences between these model results. Both models give results for the source regions and convective age of air that are broadly consistent with the measurements. Due to the numerous uncertainties, there are, however, also some results which seem to be less plausible. However, OMI $NO_2$, which is shown as proxy for boundary layer pollution, does not account for biogenic sources and is shown as average over 14 days prior to the measurement (see Sec. 2.4). Due to these limitations, additional pollution sources may have been overlooked in this analysis. Still, similar origins of highly polluted air masses, indicated by two independent backward trajectory models, agree with enhanced surface pollution, measured by OMI. This agreement within the anticipated accuracy of the two backward trajectory models suggests that both models use reliable schemes for convection detection.*

**P15, L25-26: The EMAC HNO3 mixing ratios at the tropopause look quite a bit smaller than 0.75 ppbv to me. In addition, the writing in this sentence is very awkward; I suggest rewriting as: "... flight; they decrease to values of 0.75 ppbv at the tropopause. Simulated maximum stratospheric values are not always as high as those measured, but they agree to within ... ."**
This is correct! We checked again in the data and at the tropopause in the second part of the flight, $HNO_3$ actually goes down to 0.5 ppbv. We changed the formulation according to the referee's suggestion.

**P15, L31-P16, L2: The authors posit that the diagonal feature in the HNO3 field simulated by EMAC may originate from reactions with NO2, and the tone of the discussion seems to suggest that this may be a model artifact, especially in the latter portion of the flight. But they have made no attempt, here or in the previous section, to account for the similar feature seen in the GLORIA measurements in the first half of the flight. What is the explanation for the observed structure in HNO3?**
We added a short discussion about this diagonal feature in $HNO_3$: *The difference in this diagonal structure between GLORIA and EMAC in the second part of the flight may result from a spatial displacement of the whole structure in the model, which is, however, unlikely due to the agreement of this $HNO_3$ structure in the first part of the flight, and due to the agreement of structures in pollution trace gases in the second part of the flight (see below). It is more likely that EMAC overestimates the production of or underestimates the loss of $HNO_3$ here at altitudes below 14 km.*

**P16, L5: It would be appropriate to include a reference for the CAMS assimilation of O3.**
We added the Inness et al., 2015 reference (10.5194/acp-15-5275-2015). It describes the assimilation scheme for the MACC data product, a precursor of CAMS.

**P17, L15: In addition to pointing out that the EMAC C2H2 enhancement is in the same geolocation as the measured enhancement, it would be good to note that the simulated enhancement is much weaker and less extensive than the measured enhancement; it would also be helpful to add "(cyan box)" here.**
We changed the sentence to: *In the second part of the flight, again a very small enhancement at the tropopause at 6:00 UTC (cyan box) is visible in EMAC, which is at the same geolocation as the enhancement in the measurements, but much weaker and less extensive.*

**P17, L12-15: A point that is missing from the C2H2 discussion is the fact that EMAC completely fails to simulate the maxima in the red and orange boxes and the minimum in the magenta box, even in a relative sense.**
We added the sentence: *Measured maxima of $C_2H_2$ are not reproduced by the EMAC model.*

**P17, L25-26: The authors state that their results indicate that the meteorological fields used to prescribe transport in the simulations do not include processes relevant for the observed situation. I presume that they are referring to deep convection, which is not resolved by the reanalyses, but that should be clarified. I am wondering, however, why this would be a factor only for the first part of the flight (which the sentence in question is about). According to Figure 3, as well as much of the discussion over the preceding pages of the manuscript, the second half of the flight was influenced by convection up to 150 hPa to a similar degree.**
We agree that the paragraph this sentence originates from was badly formulated. We restructured the whole paragraph and tried to be more precisely.

**P18, L57: The writing in these sentences is clumsy.  Moreover, I'm afraid**

[Figure]

**that I don't follow the logic of the arguments. First, as mentioned in an earlier comment, both portions of the flight are characterized by high convection probabilities up to 150 hPa, so for that reason alone it doesn't make sense to focus only on the second half. Second, the authors appear to be saying that \*because\* the first part of the flight is strongly influenced by convection, the simulated results would not be affected by increased emissions. But that seems backwards to me – in the absence of convection, the strength of the surface emissions would be of little consequence. This discussion needs to be clarified.**
Also based on the feedback from referee 2, we decided to move the sensitivity test that is discussed in Sec. 6 to the Supplementary Materials. We only provide a short summary of the quite lengthy discussion of Sec. 6 at the end of Sec. 5. For this reason, the sentences that are addressed by this comment are no longer part of the revised manuscript.

**P19, L6-8: I'm not sure that it is true that GLORIA did not observe the slight enhancement in HCOOH at 6:00 UTC and 16 km. There may be a faint hint of this structure in the data. Perhaps this feature should have been introduced earlier in the discussion, e.g., P17, L16-22.**
We added to Sec. 5: *In the averaged GLORIA cross sections, a small local maximum of 60 pptv is visible after 6:00 UTC, which coincides with the small enhancement in EMAC.*
However, the sentences that are addressed by this comment are no longer part of the revised manuscript.

**P19, L10-14: Of course, although the increased emissions led to larger maximum values of PAN that matched the observed peak abundances better, they did nothing to improve the structure of the simulated field. I do not think that this is an unanticipated result. I would have expected background abundances of these tropospheric tracers to rise along with peak abundances in**

**the increased-emissions scenario. So I am slightly puzzled by the discussion in these lines, which focuses on the impact of vertical resolution on the modeled fields. Its placement in this paragraph seems to imply that the smoothing effect of the coarser resolution of EMAC, which blunts peak abundances and blurs or erases fine-scale features, is somehow responsible for the background values of PAN being too high in this sensitivity test. In fact, I think that the resolution issue is just as relevant for the baseline model run, in which background abundances were also overestimated, and it would be more appropriate to move the discussion about it to Section 5.**
We added the discussion of overestimated background VMRs and the resolution issue to Sec. 5. However, the sentences that are addressed by this comment are no longer part of the revised manuscript.

**P19, L16: The possibility that model/measurement discrepancies may be partly attributable to emission sources not represented in the inventory used in these EMAC runs is mentioned. As I noted in connection with Section 2.3.1, which emission inventories were used in these simulations is a critical piece of information that has been omitted from the manuscript.**
We added the missing information to Section 2.3.1. However, the sentences that are addressed by this comment are no longer part of the revised manuscript.

**P19, L18: That the meteorological reanalyses do not resolve local deep convection is a wellknown issue that is presented here as a finding of this study. In addition, another aspect (besides the reanalyses) that does not appear to have been considered by the authors is the convective parameterization being used for these EMAC simulations. The choice of which convective parameterization is used has been shown to have a substantial impact on modeled trace gas distributions.**
For convection, we use the parameterization introduced by Tiedtke (1989) with

modifications by Nordeng (1994) as described in Tost et al. (2006). So far, this has been the best choice for our EMAC simulations. We will add this information to the EMAC description.

In addition, in the discussion of Sec. 5, we now refer to the large uncertainties of convection parameterizations used by EMAC, as reported by Tost et al. (2006).

**P19, L23: In my opinion, the statement that this study discusses "the first measurements of HNO3, O3, PAN, C2H2, and HCOOH in the center of the AMA UTLS" is too broad. While that may be true for some species of the species listed, it is not true for all of them. This statement should be qualified in some way, e.g.: first airborne measurements, or first measurements by GLORIA.**
We formulated more precisely: *This study discusses the first simultaneous airborne measurements of HNO$_3$, O$_3$, PAN, C$_2$H$_2$, and HCOOH in high spatial resolution in the center of the AMA UTLS.*

**P19, L29-30: This study is not the first to show that PAN is efficiently transported to the UTLS by deep convection, as is implied by the wording in these lines.**
We changed the sentence to: *These measurements and their analysis confirm that PAN, a precursor of O$_3$, is efficiently transported upwards by convection, and transported for a longer time in the tropopause region, as shown earlier by Glatthor et al. (2007), Fadnavis et al. (2015), and Ungermann et al. (2016).*

**P20, L6-8: Some of the discussion here is appearing for the first time in this manuscript. I do not think that it is appropriate to introduce new concepts in a section entitled "Conclusions".**
We moved this sentence to Sec. 3 and referenced this thought only briefly here.

**P20, L23-25: As noted earlier, the fact that EMAC overestimates tropospheric background mixing ratios is not unique to the increased-emissions scenario – it**

**was also the case for the baseline run, and increased emissions are expected to affect background as well as peak abundances. The same comment regarding vertical resolution applies here as well.**

We added the aspects of overestimated background VMRs and resolution to the discussion of the baseline run, while the original sentence has been omitted in the revision of the manuscript.

**P20, L31-34: These sentences are poorly written. "enhancements" are not transported upward – pollution is transported upward, leading to enhancements in the UTLS. Likewise, a "region" is not transported "around the tropopause" – the measured air masses in that region are transported. And I'm not sure what is meant by "around the tropopause".**

We changed the sentence to: *Some pollutants have been transported into the upper troposphere by convection within days before the measurements, while one part of the observed air masses remained at UTLS altitudes for a longer time.*

---

## Author Comment (AC2) · 22 Sep 2020

We thank the referee for valuable comments and suggestions. We changed all minor language and wording corrections according to the suggestions without listing all of the changes in this answer. Instead, a latexdiff document that tracks all changes made in the revised manuscript is provided in the author's response file.

To our knowledge, Copernicus will have the manuscript copy-edited by a professional writer in case of acceptance and before publication in ACP. In addition, translation services at our institution (KIT) checked the language of the revised manuscript.

[Figure]

Our answers are given below. The original referee comment is repeated in **bold**, changes in the manuscript text are printed in *italics*.

**P2, L16-17, what's the name of the aircraft campaign?**
We added the name of the Earth System Model Validation campaign to the manuscript. For the next sentence, we also mention now the Oxidation Mechanism Observations campaign.

**P2, L22-24. The logical connections of these two sentences and the previous section seems to be amiss. What's the relationship between radiative heating rates and transport in reanalyses with trace gases? Observations are sparse, so what? How do observations help? The last sentence in the next paragraph (atmospheric chemistry models ...) is out of place. It fits much better in this paragraph, instead.**
According to the comments of both referees, we reformulated and restructured this paragraph into two paragraphs: The second paragraph of the introduction now mentions studies about the transport in the ASM, focusing on open issues of vertical transport. The third paragraph of the introduction now summarizes studies of pollution trace gas measurements (and their implications) in the ASM.
In addition, we removed the "atmospheric chemistry models" sentence from the next paragraph.

**P2, L24-28. These three sentences seem to repeat themselves in various ways. It can be easily condensed into a single sentence but capture all essential elements. Please revise.**
We agree that the third sentence did not provide much new information. It would be possible to also merge the first two sentences, but in our opinion, this would not improve readability. We propose to change the first four sentences to these two sentences: *The first observations of the upper tropospheric chemical composition*

*in the region of the ASM in high vertical resolution have been obtained during the high-altitude airborne StratoClim (Stratospheric and upper tropospheric processes for better climate predictions) campaign. This study presents a unique data set of pollution trace gases, in particular non-methane volatile organic compounds (NMVOCs) obtained with the Gimballed Limb Observer for Radiance Imaging of the Atmosphere (GLORIA) during this StratoClim campaign based in Kathmandu, Nepal, 2017.*

**P3, L1. Which two models? EMAC and CAMS? Please specify. And I am sure this is not the "first evaluation" of these two atmospheric chemistry models.**
We changed the sentence to: *Second, a first evaluation in high spatial resolution in the ASM UTLS of the atmospheric chemistry models EMAC and CAMS is provided [...]*

**P3, L30. WMO, 2019 − > This one is actually "WMO, 2018"**
You are right that the WMO report referenced here is named "Scientific assessment of ozone depletion: 2018", but it was published in January 2019 (see https://www.esrl. noaa.gov/csl/assessments/ozone/2018/downloads/2018OzoneAssessment.pdf, page ii), which makes it a publication of the year 2019. However, in case of publication in ACP, Copernicus Publications will adjust the references according to their standards.

**P4, L1. Catalytic reactions with nitrogen oxides is a sink of ozone in the stratosphere and to some extent in the upper troposphere. Primary sources of ozone is the troposphere is in situ production of NOx +HOx, and NOx + peroxide radicals from VOC oxidation, hence indicator of polluted air masses.**
We refined our explanation of tropospheric ozone sources according to your comment and restructured the section according to the comment in Michelle Santee's review. In addition, we added reactions with $NO_x$ to the loss processes (Bozem et al., 2007; 10.5194/acp-17-10565-2017) and mentioned lightning as a source of $NO_x$.

**P4, L27. Maximum tropospheric mixing ratios of a few ppt for C2H2? Are you sure it is not a few ppb?? Check Xiao et al., (2007).**
Yes, you are right, that was a typo.

**P7, L14. Can you describe what are the NMVOC emission sources used in EMAC? When you say 50% and 100% additional emissions, do you mean from all emission sources, e.g. biomass burning, biofuel, fossil fuel, etc., and globally or just over Asia?**
There are anthropogenic emissions sources from biomass burning, agricultural waste burning, fossil fuels, ship, road and aircraft emissions, as well as biogenic emissions. We included this in the paper. The emissions are from MACCity, ACCMIP and RCP6.0. In our sensitivity studies, we added 50% or 100%, respectively to all NMVOC emission sources globally. We explain this now also in the description of the EMAC model simulations.

**P8, L3-4. I find this sentence very awkward, with no clear description of what was actually done.**
We changed this sentence to: *A trajectory is considered to be influenced by convection if it encounters a convective cloud during its advection, with a pressure higher than the cloud top pressure (similarly done by Tissier et al.,2016). The location of cloud encounter is then identified as a convective source.*

**P7-8. In sections 2.3, could you also provide the details on which year, time period of the model simulations that were conducted?**
For EMAC, it is already mentioned at the end of Sec. 2.3.1 that all model runs were initialized on 1 May 2017. The CAMS reanalysis is an operational data product from ECMWF and similarly to other reanalyses continuously updated. We added to the manuscript: *CAMS reanalysis data is available for the time between 2003 and 2018.* Both backward trajectory models simulate the time before the measurement. For

[Figure]

TRACZILLA, it is already stated in Sec. 2.3.3 that they are simulated for one month before the measurement. For ATLAS, we added: *Trajectories are calculated for 30 days prior to the measurement.*

**P8. Section 2.3.5 is out of place. This is observations and it should be listed in Section 2.1 or Section 2.2, not in the modeling subsection.**
Thanks for pointing that out! We modified the structure so that the OMI part is now in its own subsection.

**P9, L3 and Figure 2. I think it is more accurate to say these are colored boxes are "air masses" of interest, rather than "regions" of interest. Also in figure 2 caption, add "shows" after "the green line". I find the green line very hard to see. A thick solid dark gray line would be much better. It also distinguishes its functionality from the color boxes.**
We agree that the formulation "air masses of interest" is more accurate and changed this term throughout the manuscript. We also followed your suggestion to change the color of the 380 K tropopause to dark gray and thickened the line.

**P10, Figure 2. I find all panels very noisy, which is not surprising due to the large errors in GLORIA measurements as listed in Table 1. I would suggest average the measurement samples to larger temporal and vertical bins. This way you can average down the noise and illustrate the discussed features much better. In the present form, these features are barely distinctive from the surrounding background air masses. This is particularly problematic from C2H2 and HCOOH.**
One goal of this manuscript is to publish the GLORIA data set for this StratoClim science flight, characterized by estimated error and vertical resolution. For this reason, we prefer to present the data in the full spatial resolution. Later in the manuscript, for comparisons with the EMAC and CAMS models, we horizontally average GLORIA

profiles for a better comparison of the major structures. This averaging also reduces the noise error, which is a large contribution but not the total estimated error (see Suppl. Figs. 1, 3, 5, 7, 9). The other major contribution to the total error is the pointing error, which is not a statistical error and is thus not reduced by averaging of profiles. Vertical averaging, as suggested by the referee, would make the vertical resolution, which is an important characteristic of the retrieval, very difficult to interpret by the reader. As a compromise to better illustrate the discussed features, we changed the colorbars to discrete values instead of a continuous color spectrum, which also reduces noisy structures that are in the order of magnitude of the total estimated errors.

**P12, Figure 3 and the corresponding discussion. (a) In the text, the relevant discussion uses km as a unit while the y-axis only shows pressure. Please add the corresponding km on y-axis. (b) The cyan box in TRACZILLA show likely convective influences while ATLAS shows none. Why the two models are showing such different results? And how can the GLORIA measurements help in assessing which back trajectory model is more accurate. Also, overall, I can see TRACZILLA shows more convective influences that ATLAS. How can you assess which one is more accurate?**

(a) We followed the suggestion of the referee and added an additional y-axis with an approximation of altitude to the plots. In addition, we also mentioned the corresponding pressures in the text.

(b) "The cyan box in TRACZILLA show likely convective influences while ATLAS shows none. Why the two models are showing such different results?"

After the review, we noticed that there was a missing line in the analysis code. The enhancement in the TRACZILLA cyan box was corresponding to trajectories that were leaving the meteorological domain at higher altitudes, and they were not actually associated with convection. After applying the correct analysis, the structures of convective influence are very similar between the two models.

"Also, overall, I can see TRACZILLA shows more convective influences that ATLAS.

How can you assess which one is more accurate?"
The differences in the intensity of the convective influences between ATLAS and TRACZILLA are expected and are related to the different approaches used in the two methods, both relying on a different set of assumptions.

In ATLAS, the convective influence is estimated from the modeled detrainment rates of ECMWF ERA5 with a stochastic approach.

In TRACZILLA, the convective influence is estimated from the satellite-observed thick and high clouds. This has the advantage to give an observation-based information on the convective events based on a high temporal and spatial resolution (15 and 20 minutes and 3 and 2 km for MSG1 and Himawari data, respectively), reducing the spatial and temporal uncertainties of their identification with respect to the model-based approach. However, there are uncertainties in the determination of cloud altitudes using only passive sensors, and we have no information on the amount of mixing with ambient air at the point where the backward trajectories hits the cloud.

To conclude, ATLAS and TRACZILLA use different methods relying on different data sets (cloud satellite measurements and ERA5 detrainment rates), which have different temporal and spatial resolutions. It is expected to have different results from these different models. However, it is out of the scope of the paper to compare the two models and, in addition, it would be difficult to assess the performances of the two models from this specific case study. This issue is instead more extensively treated in the following papers: Bucci et al. (2020; 10.5194/acp-2019-1053), Legras and Bucci (2019; 10.5194/acp-2019-1075), and Wohltmann et al. (2019; 10.5194/gmd-12-4387-2019). The intent of this manuscript is to exploit the complementary information from the two models to interpret the data.

We added to the manuscript: *The different absolute percentages for convection probability for ATLAS and TRACZILLA are likely the result of the different underlying data sets and different methods for detection of convection along the backward trajectories by the models.*

**P14-P15, the discussion on various air mass signatures. For clarity and easy-to-follow purposes, I highly recommend you assemble all this information into a table. In the table, please list the type of targeting air masses, altitude at which they are sampled, surface regions where they were originated from, average measured HNO3, O3, PAN, C2H2, HCOOH concentrations within these colored boxed, transport time since they left the surface, etc. Second, please add a summary discussion on the different chemical signature of airmasses from different regions, e.g. the purple/blue box air from the marine background vs. the orange/red box air from China, etc.**

(1) We followed the suggestion of the referee and added such a table to the manuscript.
(2) We expanded the discussion at the end of Sec. 4.

**Sections 5 & 6. I found the observation-model comparison and evaluation a major weakness of this study. Neither CAMS nor EMAC produces well the observed features and gradients of all five species.This is particularly the problem for C2H2 and HCOOH. I also have problems with the brutal way of increasing NMVOC emissions by 50% and 100%. I don't see any improvements in model performance with such approach. By matching with observations better in a few patchy spots, you are also creating huge biases in other places (Figure 6) for all three species. PAN, C2H2 and HCOOH can be emitted and/or formed from various sources, i.e. anthropogenic emissions and biomass burning emissions being the highly relevant sources. The differences in the regional distributions of these sources can have a dominant impact on tropospheric distribution of these gases after they are being lofted and formed during transport. A proper way to address this model bias is to adjust the emission strength of these individual sources in separate runs and assess how do the resulted distribution change. This way, one can potentially assess the sources of these biases. You are only presenting analysis of one single flight. Therefore, such model sensitivity simulations can be easily conducted within a few days. The new**

**model results and the corresponding discussion should be included in the revised manuscript before the paper is moving forward for publication.**

We agree that the sensitivity test performed with EMAC is a very simple analysis, based on the findings by Monks et al. (2018), and not a full sensitivity study. Unfortunately, we do not share the referee's optimism to perform a full sensitivity study "easily [...] within a few days", given our workforce and computational resources. Even though we only compare and discuss one flight, every sensitivity simulation needs to include the time of at least several weeks before the measurement, in order to account for the transport of pollution from the boundary layer to the upper troposphere, where the measurements are performed.

For that reason, we decided to remove Sec. 6 from our manuscript, show results of the simple sensitivity test with +50% increased NMVOC emissions in the supplementary materials, and only briefly summarize the results of this test in Sec. 5. We think that the comparisons of GLORIA measurements to CAMS and EMAC without a detailed sensitivity study are of value on their own, as these comparisons point out considerable weaknesses of well-known atmospheric models in the upper troposphere of the ASM. In our opinion, it is also important to document negative outcomes of model evaluation studies, in order to motivate further sensitivity studies and model improvements.